# Translational reprogramming as a driver of antimony-drug resistance in *Leishmania*

Sneider Alexander Gutierrez Guarnizo [1,2], Elena B. Tikhonova[1], Andrey L. Karamyshev [1,4] ✉, Carlos E. Muskus [2,4] ✉ & Zemfira N. Karamysheva [3,4] ✉

*Leishmania* is a unicellular protozoan that has a limited transcriptional control and mostly uses post-transcriptional regulation of gene expression, although the molecular mechanisms of the process are still poorly understood. Treatments of leishmaniasis, pathologies associated with *Leishmania* infections, are limited due to drug resistance. Here, we report dramatic differences in mRNA translation in antimony drug-resistant and sensitive strains at the full translatome level. The major differences (2431 differentially translated transcripts) were demonstrated in the absence of the drug pressure supporting that complex preemptive adaptations are needed to efficiently compensate for the loss of biological fitness once they are exposed to the antimony. In contrast, drug-resistant parasites exposed to antimony activated a highly selective translation of only 156 transcripts. This selective mRNA translation is associated with surface protein rearrangement, optimized energy metabolism, amastins upregulation, and improved antioxidant response. We propose a novel model that establishes translational control as a major driver of antimony-resistant phenotypes in *Leishmania*.

Leishmaniasis is a neglected tropical disease caused by protozoan parasites of the genus *Leishmania*. This infection remains a grave public health dilemma, causing ~0.7–1 million new cases per year, with 350 million people at risk of infection[1]. Depending on the host immune system status and *Leishmania* species/strains involved, the disease can take four different clinical forms: cutaneous leishmaniasis (CL), which causes local or diffuse skin ulcers; mucocutaneous leishmaniasis (ML) that affects the mouth and nasal mucosa; visceral leishmaniasis (VL), a systemic infection that affects internal organs containing macrophages; and post-kala-azar dermal leishmaniasis (PKDL), a dermal sequela of VL. VL is lethal without treatment, causing between 20,000 and 30,000 deaths per year[2].

In the absence of an approved vaccine, the treatment for leishmaniasis relies on chemotherapy. Pentavalent antimonials, commercially administered as sodium stibogluconate (Pentostam) and meglumine antimoniate (Glucantime), have been used as the drug of choice to treat leishmaniasis during the last seven decades[3]. Nonetheless, the efficacy of antimonials is progressively diminishing, resulting in treatment failure rates of 65% in Bihar, India[4-6]. The phenomenon of resistance to pentavalent antimonials has been observed in different regions, including Asia, North Africa, and Latin America[4-8]. Even though some treatment failure could partially be explained by the host's immune system and pharmacodynamic or pharmacokinetic factors, the major loss of antimonial efficacy is consistently associated with the parasite's ability to develop drug resistance[9]. With limited antileishmanial alternatives, there is an urgent need to understand antimonial drug-resistance mechanisms.

There are indications that resistance to antimony is a complex process that involves changes in the genome, metabolome, and lipid remodeling[10-13]. Antimony-resistant parasites have been shown to

[1]Department of Cell Biology and Biochemistry, Texas Tech University Health Sciences Center, Lubbock, TX 79430, USA. [2]Programa de Estudio y Control de Enfermedades Tropicales, Universidad de Antioquia. Medellín, Medellín 050010, Colombia. [3]Department of Biological Sciences, Texas Tech University, Lubbock, TX 79409, USA. [4]These authors contributed equally: Andrey L. Karamyshev, Carlos E. Muskus, Zemfira N. Karamysheva. ✉e-mail: andrey.karamyshev@ttuhsc.edu; carlos.muskus@udea.edu.co; zemfira.karamysheva@ttu.edu

combat the drug in several ways: reducing drug uptake by down-regulation of the aquaglyceroporin 1 transporter; optimizing the antioxidant response by thiol metabolism activation; tuning the energy metabolism to fuel the high energy demanded by the anti-oxidant response; using the thiols as drug sequesters to form thiol-metal complexes; and increasing the efflux of thiol-metal complexes via exocytosis by overexpression of ABC transporters[14–16]. Nonetheless, the molecular mechanisms orchestrating the coordinated responses required to combat antimony remain unknown. Understanding these processes is crucial to improve the design of new therapeutic strategies by developing combination therapies to rescue antimonials efficacies, and further exploring the basic biology of *Leishmania* parasites.

Previous studies have shown that antimony-resistant parasites have differential expression of ribosomal proteins and proteins involved in the regulation of mRNA translation[17–21]. Nevertheless, there is no clear association between translational control and the parasite's resistance mechanisms. Interestingly, in the absence of transcriptional control, the parasites mediate changes in gene expression at the post-transcriptional level, whereby the regulation of mRNA translation assumes a central role in influencing the parasite's phenotypes[22,23].

Recent studies indicate that *Leishmania* parasites use translational regulation to modulate their phenotype in response to environmental changes[24]. It was also demonstrated that the protein kinase CDPK1 plays a role in translation and coordinates the *Leishmania* resistance to paromomycin, an antileishmanial alternative[25]. However, these studies are mostly rare, and translational regulation of *Leishmania*'s drug resistance, especially to the main antileishmanial drugs, is still very poorly studied. In this study, we have used a translatomic approach that couples polysome profiling and deep RNA sequencing to assess the changes in the translational efficiency of antimony-resistant parasites growing with or without drug challenge.

In this work, we show that antimony-resistant parasites use translational regulation to coordinate a complex response that mediates the antimony resistance. We have discovered that antimony-resistant parasites exhibit a dramatic remodeling of the translatome, even in the absence of the drug, providing a preemptive adaptation at the level of translation. We have also provided evidence that in retaliation to the drug, a highly selective translation is activated. This response involves alteration of the expression of genes directly

required to combat the drug. Translatome analysis allows us to observe changes that would be undetectable by conventional transcriptomic analysis. Our study also reveals that putative translational modulators represent the largest group of detected gene variants, supporting the importance of translational control in the development of drug resistance. We propose a new model of the role of translational control in the coordination of antimony-resistant phenotypes.

## Results

### Development of antimony-resistant strains

The resistance to antimony is considered a limiting factor for leishmaniasis treatment; however, the parasites' mechanisms for antimony resistance are poorly understood. Since *Leishmania* lacks conventional eukaryotic transcriptional control due to its polycistronic nature, gene expression is primarily regulated by post-transcriptional mechanisms. Although the drug resistance in *Leishmania* has been studied for the past 20 years, the role of translational regulation has remained essentially neglected. In this study, we explored the potential role of translational regulation in the modulation of antimony-resistant phenotypes.

To identify changes at the translational level that are associated with antimony-resistant phenotypes, the drug-resistant strains were derived from the wild-type (WT) *Leishmania tropica* strain, one of the *Leishmania* species that can induce both cutaneous and viscerotropic leishmaniasis[26–28]. Briefly, the antimony-sensitive wild-type strain was treated with an increasing concentration of trivalent antimony (Sb$^{III}$), the active form of antimonial drugs, for six months. Moderately resistant (MR) and highly resistant (HR) strains were selected by their half-maximal effective concentration (EC50), estimated by the colorimetric MTT (3-(4,5-dimethylthiazol-2-yl)−2,5-diphenyltetrazolium bromide) cell viability assay (Fig. 1a). WT, MR, and HR strains showed significantly different EC50 values: 10.46 ± 0,65; 260.37 ± 34.93; and 631.73 ± 73.7 μg/mL of Sb$^{III}$, respectively (Fig. 1b). The curves of growth for the wild type and the derived resistant strains did not show significant differences (Supplementary Fig. 1). Consistently, the MTT viability assay showed a similar signal intensity of formazan for sensitive and resistant parasites (see left wells in the panels of Fig. 1a). In addition, to test whether the resistance phenotype was stable, we submitted the highly resistant strain to 25 consecutive passages in absence of the drug (Supplementary Fig. 2). Notably, after these

**a**

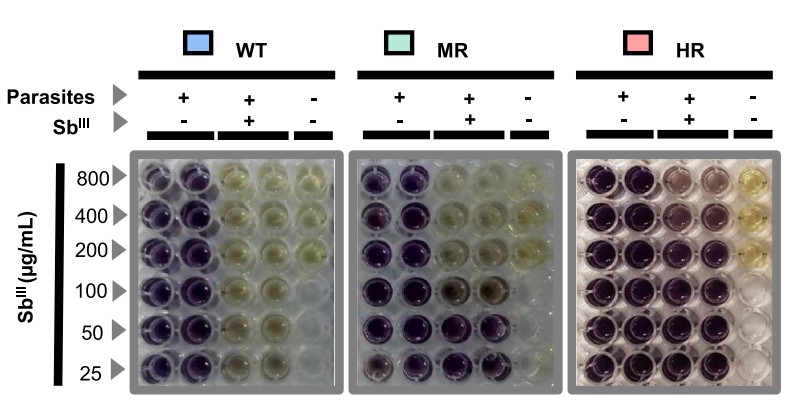

**b**

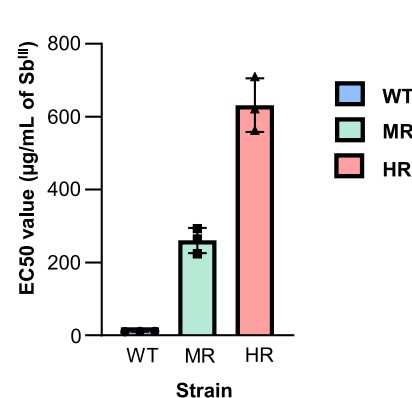

**Fig. 1 | Effective stepwise selection of Sb$^{III}$-resistant parasites.** Sb$^{III}$-sensitive or WT strain promastigotes were treated with rising Sb$^{III}$ concentration for 6 months, selecting stepwise the resistant parasites. The middle point and the endpoint of drug-resistance selection were arbitrarily considered Sb$^{III}$ moderately resistant (MR) and highly resistant (HR) strains, respectively. **a** The colorimetric MTT assay was used to estimate the EC50 value. Three conditions were evaluated per experiment: parasites growing without Sb$^{III}$ (control), parasites growing under Sb$^{III}$ pressure (treatment), and medium without parasites (blank). Negative (−) or positive (+) symbols reflect the absence/presence of parasites or drugs. The change from yellow (MTT) to purple (Formazan or reduced MTT) indicates cell viability. **b** EC50 values were calculated in three biological replicates with two technical replicates. Statistical analysis was performed by a one-way ANOVA analysis followed by Tukey's multiple comparison test. Degrees of freedom for the numerator: 2. Degrees of freedom for the denominator: 6. F distribution value: 132.2. *P* value <0.001 (**), *P* value <0.0001 (****). *P* value <0.001 (**), *P* value <0.0001 (****). *P* value (WT vs. MR): 1.54 × 10$^{-03}$. *P* value (WT vs. HR): 8.38 × 10$^{-06}$. *P* value (MR vs. HR): 1.74E × 10$^{-04}$. Data are presented as mean and ±SD. The raw data of (**b**) are available in the Source Data.

passages in culture the HR strain still showed a significantly higher EC50 value when compared to the wild type (WT: 9.34 ± 2.35; HR: 278.6 ± 80 μg/mL of Sb$^{III}$). We noticed that even though the EC50 value for the resistant strain was reduced, it still showed about 30-fold higher EC50 than the wild type. Thus, our data reveal that the HR strain has a stable resistant phenotype. An infectivity assay was completed to test whether the selected HR strain shows changes in the infection to a human phagocytic cell line in vitro. The results indicate that the derived resistant strain has a significant reduction in infectivity (Supplementary Fig. 3). As the reduction in the infectivity correlates with the level of resistance (HR versus MR and WT), it is possible that it represents parasite's trade-off adaptations to acquire drug resistance at the expense of infectivity. Consistently, various studies have previously reported a decreased infectivity in resistant parasites[29–31].

Since WT and HR strains showed clear phenotypic differences in terms of antimony susceptibility, they were chosen to conduct polysome profiling experiments followed by deep RNA-seq.

## Polysome profiling experimental design and bioinformatic workflow for translatome analysis

Upon completion of drug-resistance validation, Sb$^{III}$-sensitive and resistant *L. tropica* strains were subjected to polysome profiling followed by deep RNA-seq for comparative translatome analyses. mRNA translation is usually conducted by several ribosomes termed polysomes; more ribosomes associated with particular mRNA means mRNA is more involved in translation and likely translated more efficiently (Supplementary Fig. 4A). Polysome profiling is a technique that includes the separation of mRNA transcripts associated with a different number of ribosomes. Polysome profiling followed by deep RNA-seq is a good approach to evaluate mRNA translation status on the full translatome level. It is an adequate technique to study gene expression regulation, especially in *Leishmania* because this organism has very limited transcriptional control and its gene expression is primarily regulated during translation. The major goal of this work is to delineate the selective translation of mRNAs associated with drug resistance through analysis of mRNA engagement with ribosome and polysomes of the drug-resistant *Leishmania* strain vs WT and in response to the drug. We wanted to examine two types of changes: changes at the translational level in drug-resistant strain occurring in the absence of the drug (general basal changes), and those only detectable when the resistant parasites are growing under Sb$^{III}$ exposure (changes to combat the drug). Both of these scenarios were considered in the experimental design (Fig. 2). The highly resistant HR strain was grown with and without the drug, while the WT strain was only grown without Sb$^{III}$ because of high sensitivity to the drug; the cells were lysed, and lysates were used in polysome profiling experiments. Ribosome subunits (40 S, 60 S), monosomes (80 S), and polysomes were well separated by ultracentrifugation in sucrose gradient as demonstrated by their characteristic A$_{260}$ absorbance peaks (Supplementary Fig. 4B, C). A$_{260}$ spectra and polysome-to-monosome (P/M) ratio for WT and HR strain grown with and without the drug were very similar, indicating that the resistance to Sb$^{III}$ is not associated with a significant change in overall translational capacity (Supplementary Fig. 5 and Supplementary Data 1). A P/M mean score of 0.54 indicates that during the log phase of growth, *L. tropica* contains twice as many monosomes as polysomes, independently of the Sb$^{III}$-resistance phenotypes or drug challenge.

Total mRNAs and fractions corresponding to monosomes (one ribosome or 80 S), light (2–5 ribosomes), and heavy (6 and more ribosomes) polysomes were selected for further analysis (Supplementary Fig. 4C). mRNAs in heavy polysomes were considered to be more engaged in translation and consequently translated more efficiently than in light polysomes or monosomes. Each experimental condition was evaluated in three independent biological replicates, generating a total of 36 samples analyzed by deep RNA-seq (Fig. 2a). The produced reads were mapped against the *Leishmania major*

genome of reference (TriTrypDB-51 Friedlin strain) since it is the most completely annotated *Leishmania's* genome, and it is phylogenetically closely related to *L. tropica*. In consequence, the genes are reported using *L. major* IDs[32]. The mapped reads were used for differential translational analysis (DTA) and variant calling analysis (VCA). DTA was used to detect differentially translated transcripts (DTTs) based on the number of reads mapped per transcript and experimental condition. VCA was included to estimate the proportion of DTTs with associated gene variants. Both analyses were compared in terms of the genes and biological processes involved (Fig. 2a). The workflows for DTA and VCA were based on DESeq2 algorithm which uses negative binomial generalized linear models and the FreeBayes algorithm which uses the Bayesian statistic approach, respectively[33,34]. Both bioinformatics workflows are described in Supplementary Fig. 6.

To estimate the basal changes in the translatome after the development of drug resistance, the HR strain was compared with the WT strain (control) in the absence of the Sb$^{III}$ (Fig. 2b). To evaluate the changes in the translatome associated with active drug resistance, the HR growing under Sb$^{III}$ challenge was compared with the HR without Sb$^{III}$ treatment (control) (Fig. 2b). More specifically, monosomes were compared against monosomes, and the same type of comparison was done for light and heavy polysomes. As result, a total of six dual comparisons were included to detect DTTs (Fig. 2b). Raw and processed data are available in GEO database repository (GEO ID: GSE173848, BioProject: PRJNA727294).

Next, in order to distinguish changes in translatomes from those derived from differences in mRNA abundance, we compared the total transcriptome (total mRNA used as input for polysome profiling) and heavy polysome mRNA (more efficiently translated transcripts) as shown in Fig. 2c. DESeq2 algorithm was used for both "basal conditions" and "drug challenge". After DESeq2 analysis and the respective matching, the genes were classified into 4 groups. Group 1: genes that were detected as differentially expressed only in heavy polysomes. Group 2: genes that were detected as differentially expressed only in the total transcriptome. Group 3: genes that were detected as differentially expressed in both total transcriptome and heavy polysomes. Group 4: genes that were not considered as differentially expressed in any of the previously mentioned groups.

## Sb$^{III}$-resistant strain has a dramatically different translatome in the absence of drug challenge

To detect mRNA-specific translational changes, we first examined changes observed in the absence of drug challenge (basal level, HR compared to WT). The analysis showed that the transcript composition per polysome fraction was drastically different in the Sb$^{III}$-resistant strain even in the absence of drug (Fig. 3a and Supplementary Data 2). 49.88% were downregulated and 50.11% upregulated in total polysomes, specifically in light polysomes (906 down; 951 up), and heavy polysomes (1096 down; 1064 up). Monosomes contained only four upregulated DTTs and no downregulated ones (Fig. 3a and Supplementary Data 2). Several transcripts can be detected as a DTT in more than one polysome fraction, therefore, a Venn diagram analysis was performed to identify the distribution of unique transcripts (Fig. 3b and Supplementary Data 3). In total, 2431 differentially translated unique transcripts were detected. 1586 DTTs (65.24%) were identified in both light and heavy polysomes.

Next, we performed a comparative analysis of the total mRNA to distinguish changes in translatome from those derived from differences in mRNA abundance. The transcriptomic analysis under basal conditions demonstrates that the vast majority of changes in resistant parasites are observed in translatome (2055 transcripts—translation only, heavy polysomes) rather than transcriptome (62 transcripts—transcription only) and provides strong evidence that translational reprogramming is major driver of antimony-resistant phenotypes in *Leishmania* (Fig. 3e and Supplementary Data 4). Thus, changes in

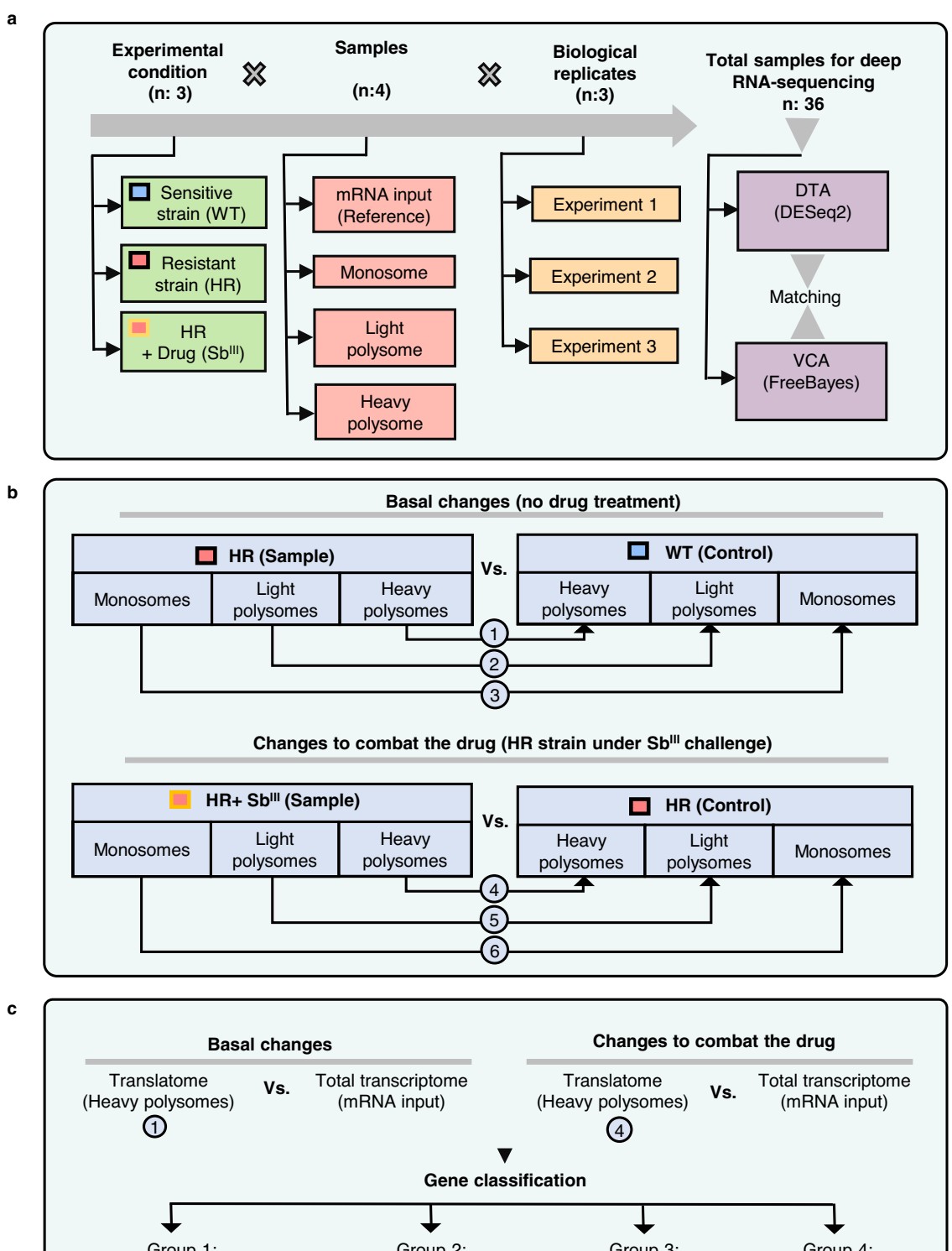

mRNAs abundance are very limited in the resistant strain, however, engagement of mRNAs in translation is drastically different from the sensitive parasites.

### The antimony challenge triggers a highly selective translation in resistant parasites

Next, we sought to determine if resistant parasites modulate the population of transcripts under translation in response to drug challenges. Compared with the dramatic changes detected at the basal level (Fig. 3), a relatively small number of DTTs was detected under the drug challenge, with the majority of DTTs exhibiting an increase (Fig. 3c and Supplementary Data 5). They were mostly associated with polysome fractions: light polysomes (up: 57, down: 9), and heavy polysomes (up: 134, down: 30), while only two DTTs were detected in monosomes (up: 2, down: 0), (Fig. 3c and Supplementary Data 5). Furthermore, the Venn diagram analysis demonstrated that the DTTs

**Fig. 2 | Schematic representation of the experimental design. a** Experimental conditions. Three experimental conditions were tested: WT and HR strains growing without drug challenge, and HR strains growing under drug challenge. Four types of samples were evaluated per experimental condition (input, monosome, light polysomes, and heavy polysomes). The experiment was done in three biologically independent replicates. A total of 36 (3X4X3 = 36) samples were used for RNA-seq followed by bioinformatic analysis. DESeq2 algorithm was used for differential translational analysis (DTA). Variant calling analysis (VCA) was used to detect gene variants exclusively present in HR strain using the FreeBayes algorithm. Then, the two bioinformatic analyses were matched based on the affected genes. **b** Detailed strategy for differential translational analysis (DTA) to estimate the basal changes in the translatome after the selection for drug resistance (basal changes), and changes associated with active drug resistance (changes to combat the drug). A total of six dual comparisons were performed including monosomes, and light and heavy polysomes. **c** Differential expression analyses were independently performed to identify changes in the total transcriptome (total mRNA used as input for polysome profiling) and translatome (heavy polysomes fraction). The identified genes were matched and classified into four groups. Group 1: genes that were detected as differentially expressed only in heavy polysomes. Group 2: genes that were detected as differentially expressed only in the total transcriptome. Group 3: genes that were detected as differentially expressed in both total transcriptome and heavy polysomes. Group 4: genes that were not differentially expressed.

correspond to 189 unique transcripts. 122 (64.5%) of these transcripts were exclusively associated with heavy polysomes, where the highest level of translational efficiency is expected (Fig. 3d and Supplementary Data 6). These results suggest that in response to the drug challenge, Sb^III-resistant parasites exhibit a shift towards a highly selective translation to prioritize the efficient translation of a specific population of mRNAs.

Remarkably, even though monosomes are the most abundant type of ribosomes based on *Leishmania*'s polysome profiles (Supplementary Figs. 1C and 2A), only a small number of DTTs were associated with them (Fig. 3a, c), and the majority of the changes in mRNAs were associated with polysomes, indicating engagement of specific mRNA population in highly efficient translation in response to the drug.

The mRNA abundance analysis identified 104 transcripts differentially expressed in transcriptome only, while analysis of heavy polysome fraction revealed 131 differentially translated transcripts (Fig. 3f and Supplementary Data 7), supporting that translational control plays a predominant role under the drug challenge. However, it appears that changes in mRNA abundance are more pronounced under the drug challenge in comparison with basal conditions (Fig. 3e, f).

### Translatome data validation by RT-qPCR and proteomic analyses

Before proceeding with the functional analysis of the translational remodeling, the translatome data were validated by both quantitative reverse transcription-polymerase chain reaction (RT-qPCR) and proteome analyses. The identification of an internal control for RT-qPCR analysis was based on the comparison of normalized read counts. Particularly, a one-way ANOVA analysis showed that mRNA of the protein corresponding to *LmjF.35.1945* in *L. major* had a similar expression level in all analyzed samples, therefore it was selected as an internal control for comparative analyses in RT-qPCR (Fig. 4a and Supplementary Data 8). In addition, OmpA was added as a synthetic mRNA Spike-In during polysome profiling fractionation as a reference for internal control identification (Fig. 4b). To simplify the validation by RT-qPCR, we focused on the changes in heavy polysomes occurring in response to the drug (Fig. 4). A total of 14 genes with different expression profiles were considered for validation and the primer design is summarized in Supplementary Data 9. A Spearman test showed a statistically significant positive correlation ($r^2$: 0.9165) between the fold changes estimated from RNA-Seq and RT-qPCR data, thus validating the translatome analysis (Fig. 4c).

Next, we asked if the DTTs detected after polysome analysis are also differentially expressed at the total RNA level (input for polysome profiling). For some gene targets, such as *TRYR* (*LmjF.05.0350*), *RING-P* (*LmjF.31.1920*), and *amastin* (*LmjF.34.1720*), the expression level was compared between total RNA input and RNA exclusively isolated from heavy polysome fractions (Fig. 4d). The analysis showed that even though some transcripts do not change at the total RNA level, they were significantly enriched in heavy polysomes in response to drug challenge, indicating a substantially increased translation despite similar total mRNA levels. Widespread uncoupling between

transcriptome and translatome is well documented, supporting that many genes are regulated at the translational level[35]. Translational control is especially important in *Leishmania* and other trypanosomatids with its polycistronic transcription[36,37], and our data support this.

Since translatome analysis detects mRNA involved in translation, we also explored whether the changes in translational efficiency are in agreement with protein levels using tandem mass spectrometry. While the DTA showed 189 DTTs in resistant parasites growing under drug challenge (Fig. 3d), the proteomic analysis identified only 56 differentially expressed proteins (DEPs). DEPs were identified by statistical filtering using the same parameters as DTTs (absolute fold change ≥1.5 and $P$ value corrected by Benjamini–Hochberg FDR ≤ 0.05). Of these DEPs, 20 were uncharacterized or hypothetical proteins. Therefore, we focused on the remaining 36 DEPs to match proteome and translatome analyses.

The majority of the proteins showed a similar expression pattern across translatome and proteome analyses. The expression pattern of 20 DEPs matched DTAs of Sb^III-resistant parasites grown under drug challenge (Fig. 4e). In addition, 6 DEPs (homocysteine S-methyltransferase, ATP-binding cassette protein subfamily A, putative calpain-like cysteine peptidase family C2, fatty acyl-CoA reductase, Sm protein F, DNA polymerase kappa) showed similar expression in the translatome of resistant parasites growing in the absence of drug challenge. The remaining 10 DEPs did not match the DTA. The comparison between proteome and translatome is summarized in Supplementary Data 10.

### The translatome analysis identifies markers associated with antimony resistance

Although the majority of the DTTs identified in response to the drug have still an unknown association with Sb^III-resistance, several of our findings are consistent with previously documented antimony resistance markers. ABC transporters are consistently linked with antimony efflux while the S-adenosylmethionine synthetase is a methyl donor generator that improves the antioxidant response[16,38,39]. These genes are frequently upregulated in Sb^III-resistant parasites and consistently were detected as more efficiently translated in response to the drug challenge in our experiments as well (Fig. 5a, c and Supplementary Data 8). The Kinetoplastid membrane protein-11, which has been reported to be downregulated in antimony-resistant parasites[40], exhibited a significant reduction in the translational efficiency in our study (Fig. 5b and Supplementary Data 8). In addition, several studies have reported the upregulation of several amastins in Sb^III-resistant parasites[41–44]. Interestingly, here we detected a group of 32 amastins displaying a dramatic increase in translation in Sb^III-resistant parasites, both with and without drug challenge (Fig. 5d and Supplementary Data 8). The shift in translatomic profiles was also validated across RT-qPCR analysis for ABC-H transporter and δ-amastin (Fig. 5e, f). Unexpectedly, the aquaglyceroporin 1 (AQP1) that modulates Sb^III uptake and is typically reported as downregulated in antimony-resistant strains[14], showed an unusual tendency of being highly translated and displayed a

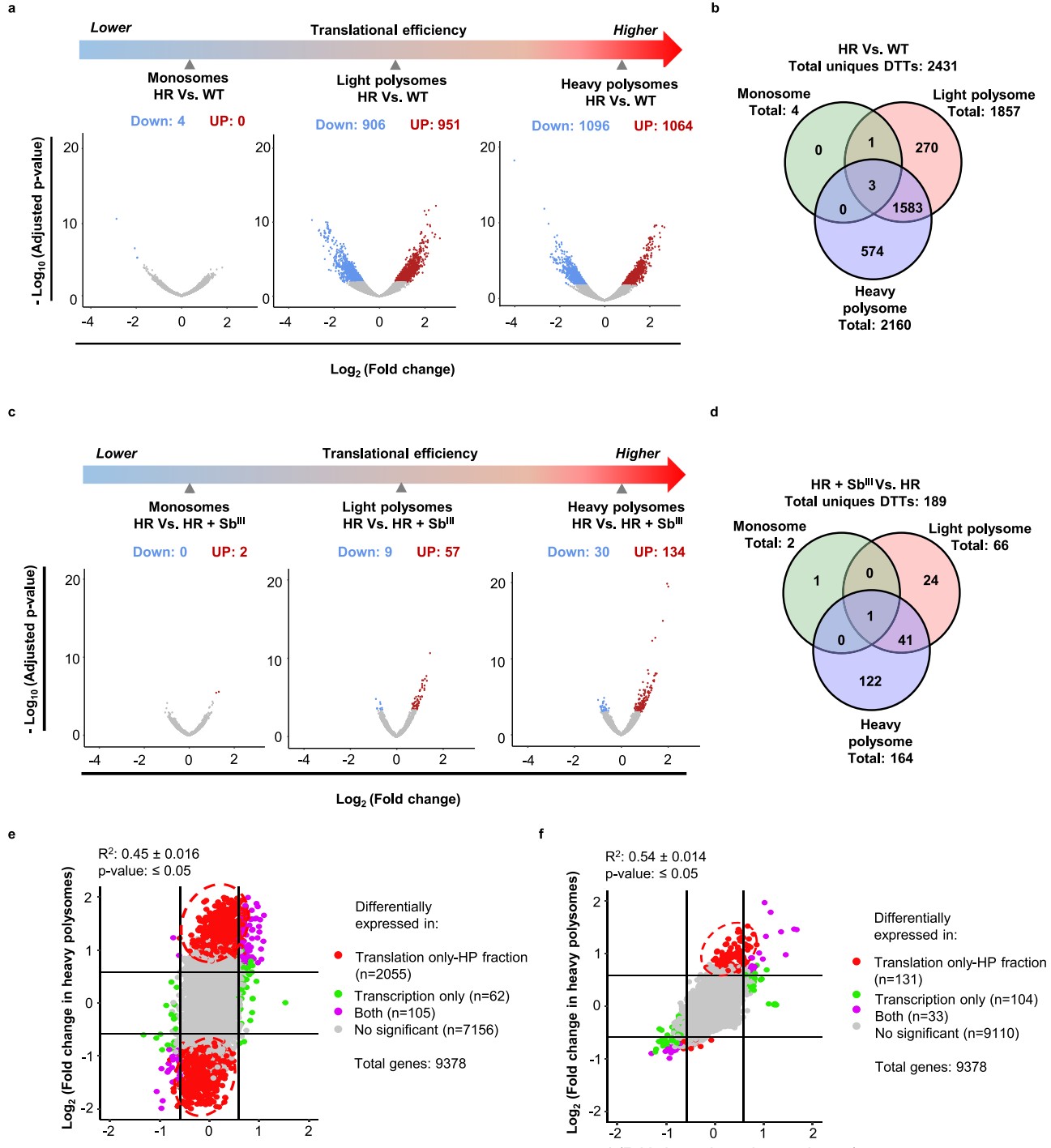

**Fig. 3 | Sb^{III}-resistant parasites exhibit dramatic translational reprogramming.** Resistant parasites show differential translatome remodeling in the absence/presence of the drug and most of the changes are detected in the translatome instead of the total transcriptome. **a** Volcano plots showing that antimony-resistant parasites growing without drug pressure have a dramatic translatome remodeling. **b** Venn diagram showing DTTs distributed per ribosome fractions in resistant parasites in comparison to sensitive in the absence of drug challenge (basal changes). **c** Volcano plots demonstrating significant changes in translational efficiency for certain transcripts in the presence of drug challenge (HR + Sb^{III} Vs HR). **d** Venn diagram showing DTTs distributed per polysome fractions after drug challenge. **e** Comparison of the translatome Vs. transcriptome at the basal level. n: three biologically independent replicates. Two-sided pearson correlation. t distribution score: 48.659, degree of freedom: 9246, $P$ value $<2.2 \times 10^{-16}$, $R^2$: 0.451, 95% IC (0.435, 0.467). **f** Comparison of the translatome Vs. transcriptome under the drug

challenge. n: three biologically independent replicates. Two-sided pearson correlation. t distribution score: 61.974, degree of freedom: 9253, $P$ value $<2.2 \times 10^{-16}$, $R^2$: 0.541, 95% IC (0.527, 0.555). **a, c** The $Y$ axes represent the adjusted $P$ values estimated by Benjamini and Hochberg method and transformed by the negative logarithm base 10 or - $Log_{10}$ (Adjusted $P$ value), higher values correspond to smaller $P$ values or more reliable. logarithm base twofold change or $Log_2$ (Fold change). Negative and positive values correspond to the downregulated and upregulated genes, respectively. Significant decrease (down). Significant increase (up). **e, f** Genes that were detected as differentially expressed only in heavy polysomes (translation only). Genes that were detected as differentially expressed only in the total transcriptome (transcription only). Genes that were detected as differentially expressed in both total transcriptome and heavy polysomes ("both"). Genes that were not considered as differentially expressed (no significant). The raw data of (**a**–**f**) are available in Source Data.

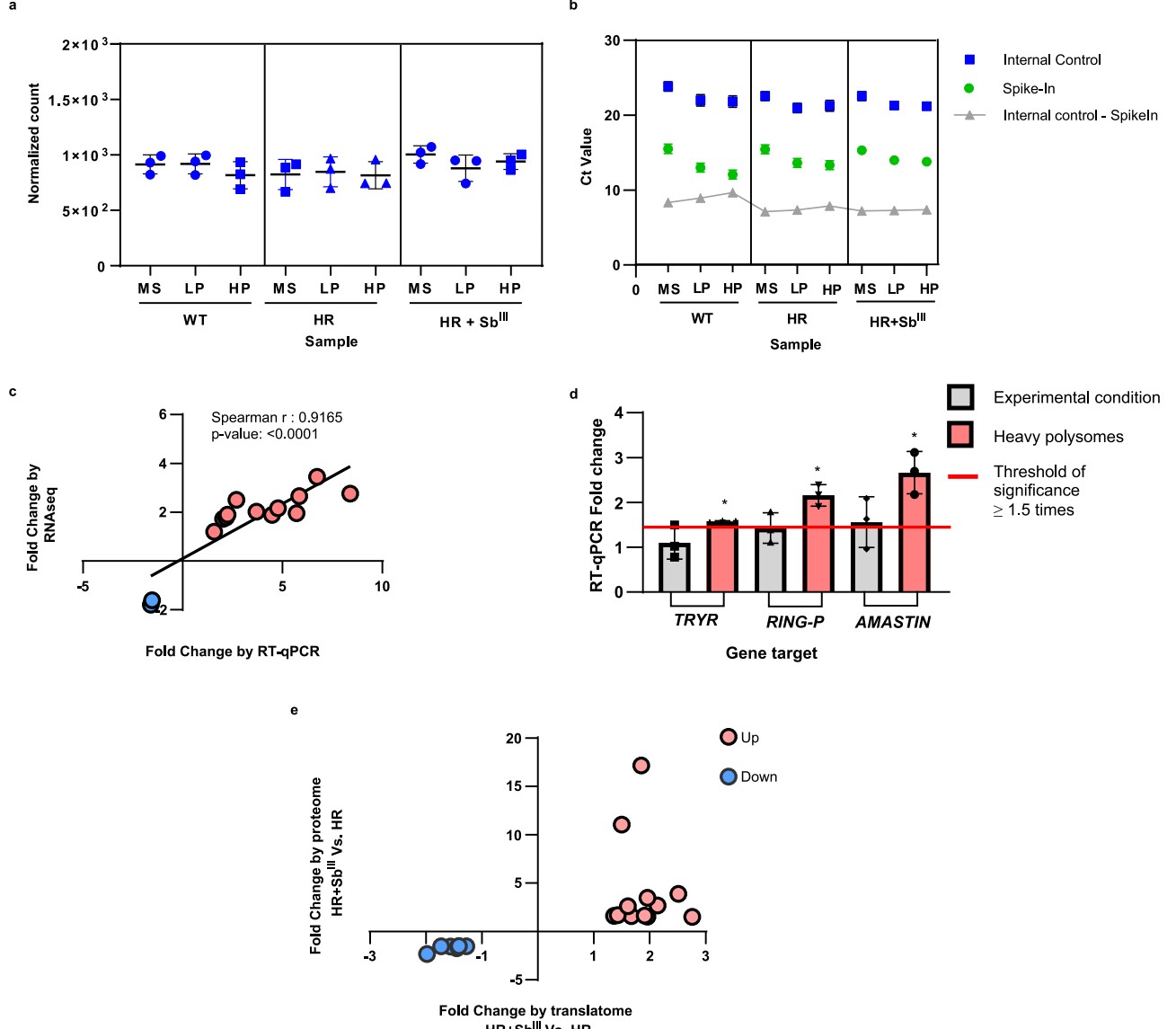

**Fig. 4 | Validation of translatome data by RT-qPCR and proteome analyses.**
**a** Internal housekeeping gene (*LmjF.35.1945*) for RT-qPCR validation showing similar expression levels in all samples. The normalized read count across all samples was compared using a one-way ANOVA analysis. n: 27 observations. Degrees of freedom for the numerator: 11. Degrees of freedom for the denominator: 24. F distribution value: 1.594. *P* value: 0.1639. Monosomes (MS), light polysomes (LP), and heavy polysomes (HP). n: 3 biologically independent replicates. Data are presented as mean values +/− SD. **b** Comparison of the internal control (IC, blue dots) and OMPA RNA used as Spike-In (green dots) showing a similar tendency in the distribution of Ct values across the experimental conditions; n: two independent biological replicates with three technical replicates. **c** Spearman analysis showing a significant positive linear correlation between the fold changes (not log-transformed) estimated by translatome analysis or RNA-Seq (*Y* axis) and RT-qPCR analysis (*X* axis). n: three independent biological replicates with three technical replicates. Two-tailed Spearman correlation. rho: 0.9165, 95%

CI: 0.7435 to 0.9745, alpha: 0.05, *P* value: $1.37 \times 10^{-5}$. The validation by RT-qPCR only included the heavy polysome fraction of resistant parasites growing under the Sb[III] challenge. n: 3 independent biological replicates with three technical replicates. **d** Comparison of the fold changes detected by RT-qPCR (*Y* axis) using two different types of samples, total RNA input (gray bar) and heavy polysome fraction (red bar). In both cases, the parasites were grown under Sb[III] challenge. Unchanged genes at the total RNA level were detected as significantly differentially translated in heavy polysomes. Threshold of significant fold change ≥1.5 (red line). Effect sizes over the cutoff of 1.5 (asterisk mark). Not significant DEG (gray bar). **e** The linear fold changes were compared between proteomic and translatomic analyses (HR + Sb[III] Vs HR). Protein/transcripts consistently upregulated (red dots). Protein/transcripts consistently downregulated (blue dots). A total of 20 proteins were matched with respective transcripts and these are described in Supplementary Data 10. The raw data of (**a**–**e**) are available in Source Data.

higher association with light polysomes in Sb[III]-resistant parasites challenged with the drug (Supplementary Data 5). The potentially new and previously reported antimony-resistant markers are summarized in Supplementary Data 11. Interestingly, the identified antimony modulators are involved in specific biological processes, such as calcium metabolism, energy metabolism, flagellar component, lipid metabolism, surface protein remodeling, translational control, vesicle transport, and drug efflux.

## Sb[III]-resistant parasites growing without drug challenge display a preemptive adaptation to antimony drug through dramatic changes in translatome

After translatome data validation, we focused on the biological interpretation of the translational remodeling and identification of key regulators potentially contributing to drug resistance. First, we concentrated on the role of the dramatic changes in the translatome observed in Sb[III]-resistant parasites growing without drug challenge.

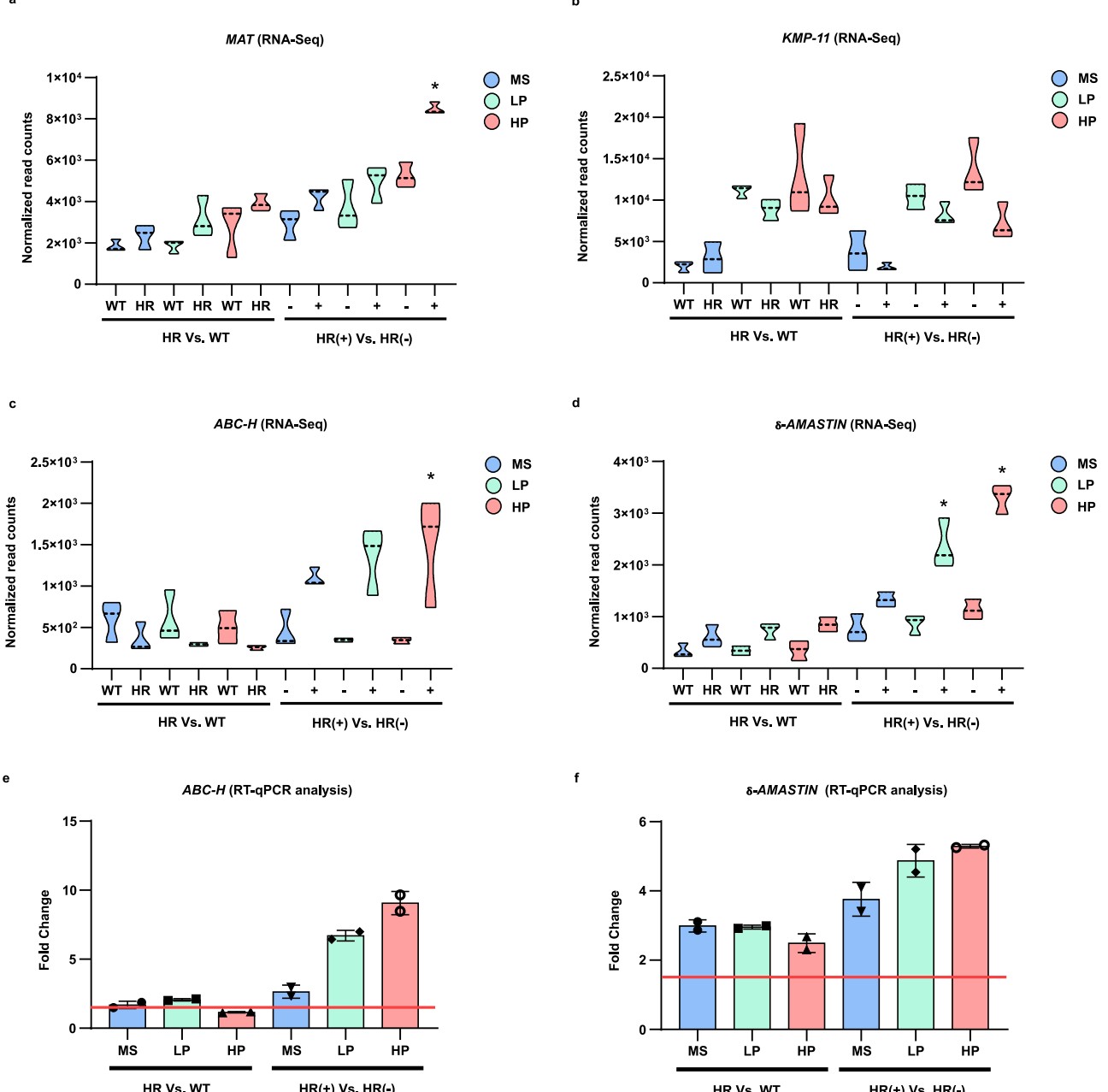

**Fig. 5 | Transcripts associated with the mechanisms of Sb^III drug resistance show a shift in translational efficiency.** RNA-Seq (normalized read count) and RT-qPCR (mRNA fold change) were used to represent the changes in translational efficiency per polysome fraction and gene target. The *Y* axis represents the read counts normalized by DESeq2's median of ratios. The *X* axis shows polysome fraction distributed per experimental condition in two panels, basal changes (left panel in each graph) and changes to mediate drug resistance (right panel in each graph). Drug-treated (+), not treated (−). **a** The S-adenosylmethionine synthetase (MAT, *LmjF.30.3500*) typically upregulated in Sb^III-resistant parasites. *P* value (MS): 0.84, *P* value (LP): 0.11, *P* value (HP): 0.009 (*). **b** The Kinetoplastid membrane protein-11 (KMP-11, *LmjF.35.2210*) is typically downregulated in Sb^III-resistant parasites. *P* value (MS): 0.76, *P* value (LP): 0.50, *P* value (HP): 0.094. **c** The ATP-binding cassette protein subfamily H (ABC-H, *LmjF.11.0040*) is typically upregulated in Sb^III-resistant parasites. *P* value (MS): 0.69, *P* value (LP): 0.06, *P* value (HP): 1.33 × 10^{−05}(*).

**d** δ-amastin (*LmjF.31.0450*) a surface protein with an unknown role in Sb^III-resistance. *P* value (MS): 0.63. *P* value (LP): 0.02 (*). *P* value (HP): 7.53 × 10^{−10} (*). **a**−**d** Two-sided Wald test. Multiple comparison corrected by the Benjamini–Hochberg Procedure. Adjusted *P* value for drug challenge condition (right side of the figure). n: three biologically independent replicates. The shift in translational efficiency was also detected by RT-qPCR analysis. **e** Expression of ABC-H as linear fold change distributed per polysome fraction and experimental condition after qPCR analysis. **f** Expression of δ-amastin as linear fold change distributed per polysome fraction and experimental condition after RT-qPCR analysis. **e**, **f** n: two independent biological replicates with three technical replicates. Data are presented as mean values +/− SD. Threshold of significant fold change ≥1.5 (horizontal red line) for RT-qPCR. Monosomes (MS), light polysomes (LP), and heavy polysomes (HP). The raw data of (**a**−**f**) are available in Source Data.

Gene ontology (GO) and metabolic pathway enrichment analysis were accompanied by protein–protein interaction networks based on both homology and co-expression to identify key modulators of Sb^III-resistance. While networks based on homology allow for the clustering of structurally and functionally related proteins, the networks based on co-expression analysis can be used to identify genes with a regulatory role in drug resistance with important therapeutic implications[45].

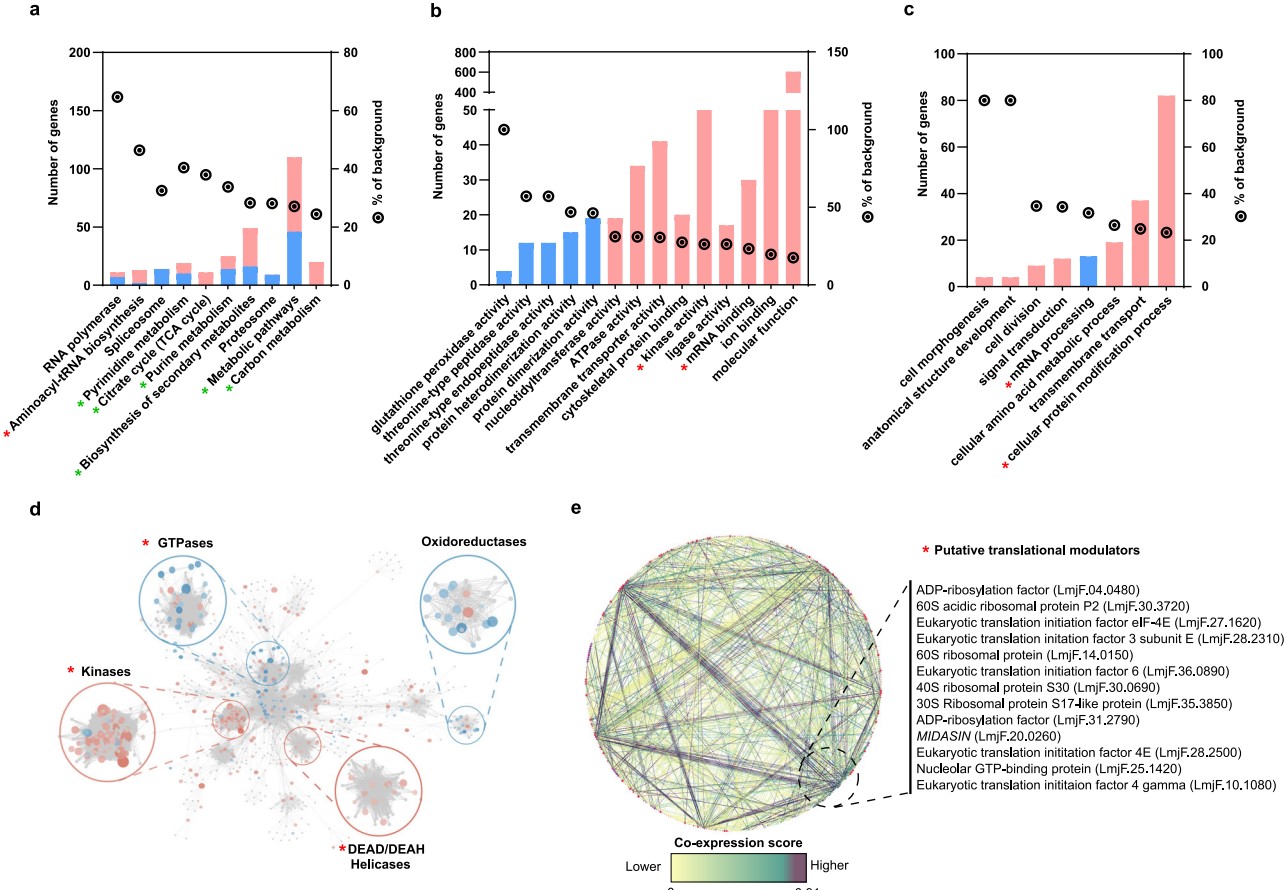

**Fig. 6 | Global translatome analysis of resistant parasites growing without drug challenge reveals complex preemptive adaptations to facilitate drug resistance.** Gene ontology (GO) and metabolic enrichment analysis were performed using the TriTrypDB browser. The number of genes per category is shown on the left of the *Y* axis. The right *Y* axis represents the percentage of the total background per category as a parameter of enrichment shown as a circle with a dot. **a** Enriched metabolic pathways. **b** Enriched molecular function GO categories. **c** Enriched biological process GO categories. **a**–**c** GO categories significantly enriched were filtered by FDR ≤ 0.05 using Benjamini−Hochberg method. Upregulated DTTs (pink), downregulated DTTs (blue). **d** Protein−protein interaction network based

on connectivity and betweenness centrality by protein structure, experimental evidence, and protein family, highlighting four enriched clusters. **e** STRING protein−protein interaction network based on known protein co-expression or putative homologs co-expressed in other organisms. A core of translational modulators can potentially regulate the expression of several components of the network based on the number of interactions and co-expression scores. Co-expression scores are shown on a scale of 0 (yellow) to 1 (dark purple), where 1 is the higher co-expression level between two proteins. Proteins potentially associated with translational control are shown on the right side. The raw data of (**a**–**e**) are available in Source Data.

The enrichment analysis by functional categories, biological processes, metabolic pathways, protein homology, and protein co-expression is shown in Fig. 6. As more than two thousand DTTs associated with polysomes were detected in resistant parasites in comparison with sensitive ones in the absence of Sb$^{III}$ (basal changes), the significantly enriched GO categories were diverse and complex. However, two types of processes were notably enriched—energy metabolism (green asterisk) and putative translational modulators (red asterisk, Fig. 6).

Several molecular functions and metabolic pathways associated with energy metabolism (Fig. 6a and Supplementary Data 12) were predominantly upregulated and therefore more efficiently translated in resistant parasites (purine metabolism, carbon metabolism, pyrimidine metabolism, biosynthesis of secondary metabolism, and cellular amino acid metabolic process). In addition, various ion-binding proteins and transmembrane transporters were also associated with an increase in drug resistance (Fig. 6b, c). Interestingly, among numerous implicated mechanisms, our analysis showed the enrichment of proteins that can potentially participate in mRNA translation, including "mRNA binding proteins" and "kinase activity" (significantly enriched GO categories associated with gene molecular function); "mRNA processing", and "aminoacyl t-RNA biosynthesis" (Fig. 6b, c and Supplementary Data 12).

Complementarily, the interactome based on protein homology showed four enriched clusters (Fig. 6d and Supplementary Data 13). Three of these clusters involve potential translational modulators. A first cluster grouped several GTPases. In this cluster, 27 DTTs were identified and most of them were decreased in resistant parasites. Interestingly, besides cytoskeleton rearrangement, GTPases can modulate different stages of protein biosynthesis by guanosine triphosphate (GTP) hydrolysis[46]. A second cluster included a noteworthy group of 62 kinases that were mostly increased in resistant parasites, several of them correspond to serine/threonine-protein kinases that can participate in multiple processes including parasite proliferation, response to stimulus, and translational regulation[47]. A third cluster grouped 15 DEAD/DEAH box helicases that were predominantly increased in resistant parasites. These proteins are typically involved in ribosome biogenesis[48–50]. Furthermore, other helicase domain-containing proteins, such as eukaryotic translation initiation factor 4E-1 (*LmjF.27.1620*, downregulated), 4E-2 (*LmjF.19.1500*, downregulated), and 4E-3 (*LmjF.28.2500*, upregulated) were also differentially translated. Finally, a fourth cluster contained mostly downregulated DTTs in resistant parasites growing without drug challenge. It included 15 oxidoreductases, mainly constituted by quinonoid dihydropteridine reductases (QDPR), enzymes that participate

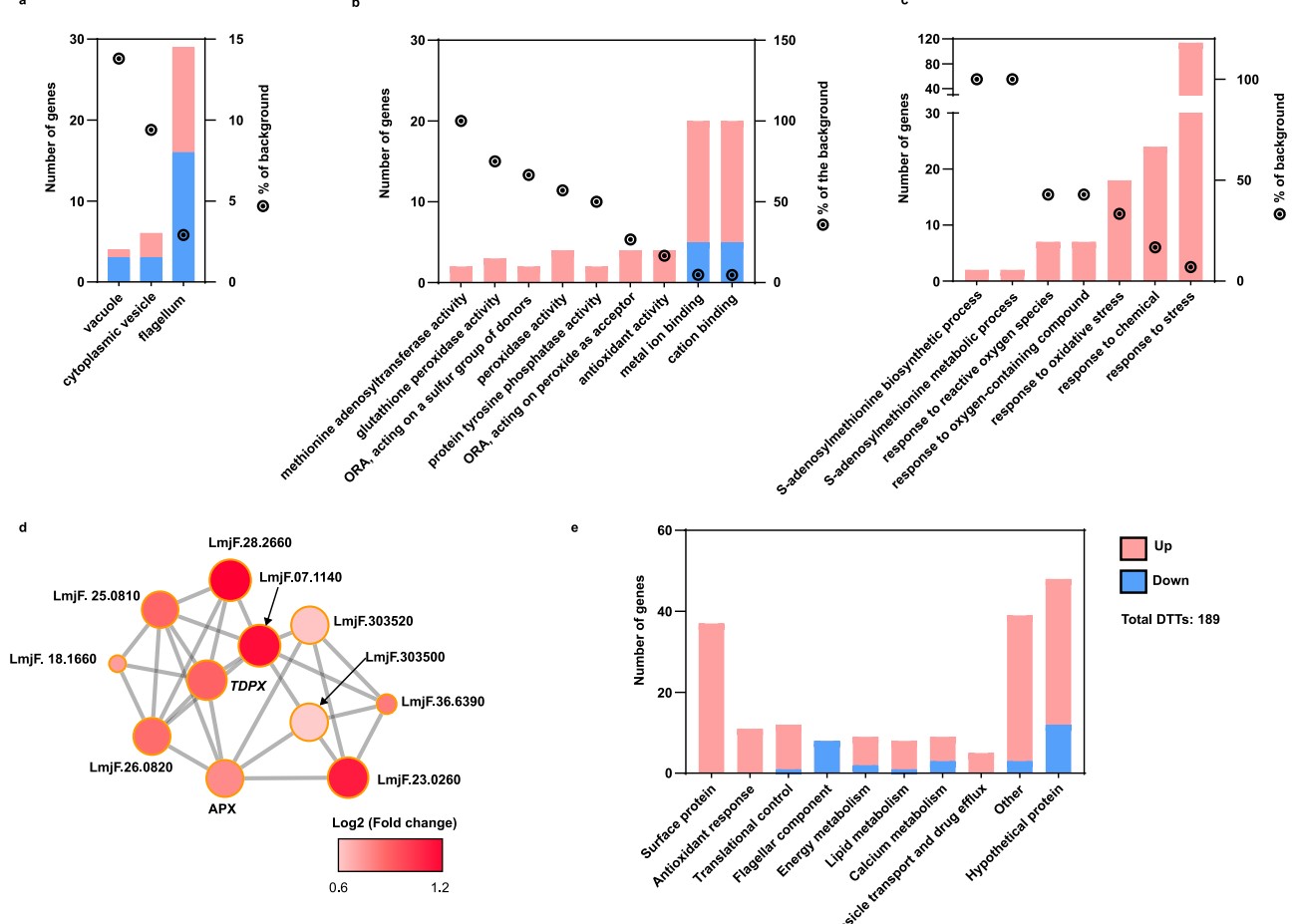

**Fig. 7 | Resistant parasites modulate the synthesis of proteins involved in interconnected processes to combat Sb^III challenge.** DTTs detected in antimony-resistant parasites growing under drug challenge were submitted to GO analysis and protein–protein interaction network. **a** Significantly enriched cellular component GO categories including cytoplasmic vesicle, flagellum, and vacuole. **b** Significantly enriched molecular function GO categories including antioxidant modulators and ion-binding proteins. **c** Significantly enriched biological process GO categories supporting the activation of antioxidant response. **a–c** Upregulated DTTs (pink-red bars and nodes). Downregulated DTTs (blue bars). GO categories

significantly enriched were filtered by FDR ≤ 0.05 using Benjamini–Hochberg method. **d** STRING protein–protein interaction network suggesting that proteins involved in glutathione/trypanothione (two molecules of glutathione joined by spermidine) metabolism are translated with high efficiency. Ascorbate peroxidase (APX, LmjF.34.0070), tryparedoxin peroxidase (TDPX, LmjF.36.0800), methionine sulfoxide reductase (METK1, LmjF.07.1140, LmjF.30.3500). **e** DTTs manually grouped based on the annotated gene description. The raw data of (**a**–**e**) are available in Source Data.

in the pathway that recycles the tetrahydrobiopterin (BH4), an antioxidant compound that mediates the $H_2O_2$ susceptibility[51]. Interestingly, the interactome based on protein co-expression showed that a group of at least 13 proteins involved in translation was significantly enriched representing multiple interactions in the network (Fig. 6e and Supplementary Data 14). Particularly, the changes in ribosomal proteins and translation initiation factors could have a key role in the observed translational remodeling, possibly as a preemptive adaptation to drug challenges.

Thus, our analysis revealed that even in the absence of the drug the translatome of resistant parasites is dramatically different from the sensitive strain. As a result, many important processes in the cell, such as energy and amino acid metabolism, transmembrane transport, mRNA binding, mRNA processing, and putative translational modulators are substantially modified.

**Drug-resistant parasites activate a highly selective translation in response to antimony drug**

As described above the resistant parasites displayed a dramatic remodeling of translatome in comparison with sensitive parasites even in the absence of drug thus supporting that preemptive adaptation is

necessary to combat the drug efficiently. Interestingly, their response to the drug challenge was very selective and included only 189 unique DTTs (Fig. 3d). To understand the biological significance of translational changes detected in Sb^III-resistant parasites in response to the drug further analysis was performed. As described before, GO and protein–protein network interaction analyses were carried out to identify key players in Sb^III-resistant parasites in response to drug challenges.

The significantly enriched GO categories associated with cellular components showed notable changes in the translational efficiency of mRNAs encoding proteins associated with microtubule organization (cytoplasmic vesicle, flagellum, and vacuoles) (Fig. 7a and Supplementary Data 12). The significantly enriched GO categories associated with molecular functions identified a highly efficient synthesis of proteins involved in antioxidant response (methionine adenosyltransferase activity, glutathione peroxidase activity, peroxidase activity) and ion-binding proteins (metal ion binding, and cation binding) (Fig. 7b and Supplementary Data 12). In addition, components of the antioxidant response were also enriched in biological process GO categories (S-adenosylmethionine metabolism, response to reactive oxygen species, response to oxidative stress, response to stress, etc.)

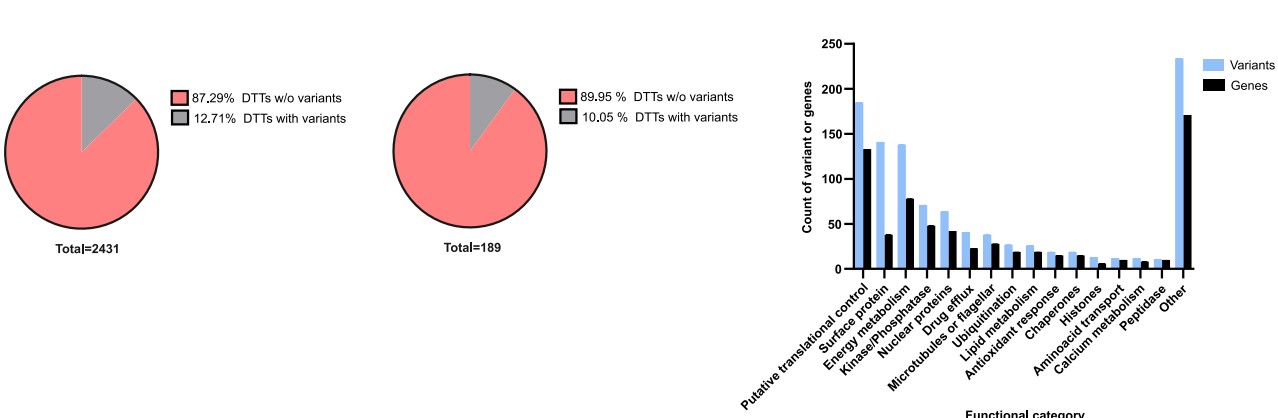

**Fig. 8 | Only a minor group of DTTs showed associated variants but the functional categories involved are consistent between DTA and VCA.** The gene variants exclusively detected in Sb^III-resistant parasites were matched with differential translational analysis and distributed per gene functional categories. **a** Pie plot summarizing the percentage of DTTs detected at the basal level with associated gene variants. **b** Pie plot summarizing the percentage of DTTs detected after drug challenge with associated gene variants. **c** Bar plot summarizing the number of variants (blue bars) and associated genes (black bars) per functional category. Functional categories were manually annotated based on the gene function. Only functional categories covering at least ten variants or genes were included. Variants associated with genes with unknown functions were not considered. The raw data of (**a**, **b**) are available in Source Data.

(Fig. 7c and Supplementary Data 15). Finally, the STRING interaction network showed significant enrichment of proteins associated with glutathione metabolism, equivalent to trypanothione metabolism of *Leishmania* parasites, the main antioxidant response in trypanosomatids. These 11 nodes are all highly interconnected (Fig. 7d and Supplementary Data 15).

Since GO analysis is limited by gene annotation, we manually grouped the DTTs detected under antimony challenge in general functional categories based on gene annotation. The DTTs with known or putative functions were grouped into 9 general functional categories which were clearly interconnected. Balanced "calcium metabolism" (9 genes) and "antioxidant response" (11 genes) can participate in mediating the oxidative stress typically induced by the antimony and promote the drug inactivation via thiol metabolism[52]. Optimized "energy metabolism" (9 genes) could supply the high-energy cost required by the antioxidant response activation and other drug depuration mechanisms[10]. Since the flagellar movement is associated with a high-energy cost, the decrease in "flagellar components" (8 genes) can contribute to optimizing energy metabolism by conserving ATP[24]. Complementary experiments showed that decreased flagellar components correlate with a smaller flagellum in resistant parasites growing under drug challenge (Supplementary Fig. 7). "Vesicle transport and drug efflux" (5 genes) can mediate drug removal[16]. The largest group included "Surface protein" (37 genes) that can modulate the parasite's membrane composition and drug transport[53]. "Lipid metabolism" can contribute to membrane remodeling and energy metabolism optimization[10]. "Translational control" (12 genes), can potentially participate in the activation of highly selective translation in response to the drug (Fig. 7e). These results support the idea that the mechanisms to combat the Sb^III require complex, coordinated processes. The DTTs manually grouped are summarized in Supplementary Data 16.

### Differential translational and variant calling analyses detected different genes involved in common biological processes

Finally, we asked if the observed translatome remodeling correlated with changes at the genomic level by the detection of gene variants exclusively observed in resistant parasites. After filtering, a total of 1626 variants were detected. Of them, 623 (38.31% of the total) were homozygous. In terms of the type of variant, most of them correspond to single nucleotide polymorphisms or SNPs ($n = 1254$ or 77.12% of the

total). Interestingly, concerning the genomic region, the variants were mostly distributed in regulatory regions, upstream of the annotated genes (1195 or 73.49% of the total). This observation is consistent with recent studies demonstrating that mutations occur less often in coding sequences of eukaryotic organisms[54]. Moreover, only 186 (11.44% of the total) missense variants were detected. Particularly, 57 missense variants were homozygous, suggesting a higher probability to impact the parasite's phenotype and have a protagonist role in Sb^III resistance (Supplementary Fig. 8 and Supplementary Data 17).

Then, we estimated the percentage of DTTs with associated gene variants to analyze the correspondence between DTA and VCA. The results showed that most of the DTTs do not contain associated variants. At the basal level, 309 DTTs (12.71% of the total of 2431) showed associated variants (Fig. 8a and Supplementary Data 18). Under the drug challenge, 19 DTTs (10.05% of the total of 189) showed associated variants, and most of them correspond to amastins (Fig. 8b and Supplementary Data 18). These results indicate that genomic changes can only partially explain the dramatic remodeling detected at the translational level. Interestingly, the manual grouping of variants by general functional categories showed that the drug-resistant strain exhibits variants in genes that are associated with functional categories that were also enriched in the translatome analyses (Figs. 7 and 8). These categories include putative translational control, surface proteins, energy metabolism, and kinase/phosphatase proteins (Fig. 8c and Supplementary Data 18). This evidence indicates that during the development of resistance, continuous drug exposure induces selective pressures on some parasite's biological processes that gradually leads to a coordinated rearrangement at both genomic and translatomic levels.

## Discussion
Leishmaniasis is a complex disease, with cutaneous, mucocutaneous, and visceral forms, that is caused by *Leishmania* parasites and is difficult to treat due to the widespread emergence of drug-resistant strains. Whereas cutaneous leishmaniasis is the most common form of the disease affecting humans, visceral leishmaniasis is the most life-threatening, ranking second to only malaria as the most fatal of parasitic infections. Drug resistance in *Leishmania* parasites remains a restrictive factor in leishmaniasis treatment; however, the basic molecular mechanisms orchestrating the development of resistance remain poorly understood. In the absence of a vaccine, leishmaniasis

disease treatment relies heavily on chemotherapy. Current chemotherapy drugs used to treat leishmaniasis are limited. These include pentavalent antimonials (Sb$^V$), as the primary treatment, and amphotericin B, miltefosine, pentamidine, and paromomycin as alternative treatments[55]. However, treatment of leishmaniasis is obstructed by drug toxicity, high cost, and treatment failures caused by drug resistance. Furthermore, cross-resistance phenomena have been documented, suggesting a high capacity of *Leishmania* to adapt to different stressors and drugs[56,57]. Resistance to antimony, the primary antileishmanial chemotherapy drug, is related to genomic rearrangements that can modulate the dosage and function of key proteins[11,43,58]. However, genomic rearrangements and mutations do not explain all possible drivers of resistance mechanisms. *Leishmania*, and other trypanosomatids, have polycistronic transcription and the use of transcriptional control is limited[59]. As a result, *Leishmania*, and other trypanosomatids, use primarily translational control to regulate gene expression[60–62]. Therefore, besides the changes at the genomic level, translational control could represent the major route to combat the drugs. Nevertheless, the role of translational regulation of drug resistance in *Leishmania* has not been investigated yet. The major advantage of translational regulation is that it can be easily reversed and readjusted under different environments[63]. This study describes a new perspective on drug resistance in *Leishmania* parasites, highlighting translational reprogramming as a crucial driver of antimony-resistant phenotypes.

In this study, a *L. tropica* strain that is highly resistant to antimony drug was generated on the base of a sensitive strain upon a gradual increase in drug concentration (Fig. 1). The changes in the translatome were studied by coupling polysome profiling and deep RNA-Seq (Supplementary Fig. 4). Two types of comparison were used to identify changes in the translatome exhibited by the derived resistant strain. First, the resistant strain was compared with the sensitive strain to identify basal translatome changes in the absence of drug challenge. Then, the translatome of resistant parasites treated with the antimony drug was compared with the translatome of untreated resistant parasites to identify potential drug-resistance modulators (Fig. 2).

Although the overall translational capacity remains unchanged (Supplementary Fig. 5) and the population of transcripts associated with monosomes remained mostly unaffected (Fig. 3), the population of efficiently translated transcripts (associated with light and heavy polysomes) was significantly different in antimony-resistant parasites under both, basal changes and in response to the drug. Notably, the basal changes of the translatome in the absence of the drug included 2431 differentially translated transcripts in comparison with sensitive parasites (Fig. 3a, b). This suggests that the development of antimony resistance involves dramatic reprogramming of mRNA translation. Our comparative analysis of transcriptome versus translatome reveals very limited changes in mRNA abundance and is in a strong favor of our model supporting that translational reprogramming plays a major role in drug resistance in *Leishmania* parasites (Figs. 3e, f and 9). The basal changes of the translatome observed in resistant parasites could represent the parasite's strategies to efficiently compensate for the biological fitness once they are exposed to the antimony (Fig. 9a). In our previous work using the same resistant and sensitive strains, we observe that lipid and metabolite composition is different in the resistant strain even in the absence of the drug correlating well with dramatic changes in translatome observed under basal conditions[10,13]. In future studies, it would be interesting to examine several resistant cell lines derived from the same parental sensitive strain and determine if parasites always use the same route of translational reprogramming to develop drug resistance or different solutions are possible to achieve the same outcome by means of translational regulation. The association between drug-resistant phenotypes and reprogrammed mRNA translation has been previously observed in refractory cancer cells. Translational reprogramming is commonly recognized as a source of adaptive plasticity that allows cancer cells to become resistant to the new therapies[45,64–66]. The identification of key translational modulators associated with drug resistance in *Leishmania* parasites is essential for the development of new therapeutic strategies.

Resistant parasites that are grown while exposed to the antimony induced changes in the translation of a specific population of only 189 transcripts. These drug-resistance modulators were grouped by interconnected biological processes, such as surface protein remodeling, improved energy metabolism, oxidant response, drug inactivation, and drug efflux (Fig. 9b). Together, our results suggest that *Leishmania* parasites use multifaceted mechanisms to combat the drug's effect in a coordinated effort. In this study, we have shown that antimony-resistant parasites optimize energy metabolism through changes in translational efficiency (Figs. 6c, 7e, and 8c). In agreement, our previous findings indicated that an optimized carbon-energy metabolism achieved through lipidome and metabolome changes is essential to fuel the high-energy demands required to cope with antioxidant-induced stress and drug efflux via ABC transporters[10,13] (Fig. 8). In addition, resistant parasites downregulate flagellar components, a strategy that could promote an optimized energy expenditure (Fig. 7a). We observed that this event correlates with the resistant parasite's phenotype of possessing a shorter flagellum, which is associated with lower flagellum activity and lower energy expenditure (Supplementary Fig. 8). Based on the increase of proteins argininosuccinate synthase (*LmjF.23.0260*), ascorbate peroxidase (*LmjF.34.0070*), gamma-glutamylcysteine synthetase (*LmjF.18.1660*), methylenetetrahydrofolate reductase, putative, peptide methionine sulfoxide reductase-like, S-adenosylmethionine synthetase (*LmjF.30.3520*, *LmjF.30.3500*), SelR domain-containing protein (*LmjF.28.2660*), tryparedoxin peroxidase (*LmjF.26.0800*, *LmjF.26.0820*), it is evident that the resistant parasites preferentially utilize trypanothione metabolism as an antioxidant response (Fig. 8d). These results have been consistently verified by several studies claiming that trypanothione metabolism is involved in drug inactivation through the formation of thiol-metal complexes, which are necessary for drug efflux[15,67–69]. We also identified proteins involved in drug efflux via exocytosis, such as ATP-binding cassette protein subfamily H (*LmjF.11.0040*), QA-SNARE (*LmjF.29.0070*, *LmjF.19.0120*), Qb-SNARE (*LmjF.19.0010*), and vesicle coat proteins (*LmjF.29.1810*)[70]. Downstream, drug efflux requires remodeling of the parasite's membrane to allow for the fusion of vesicles containing the thiol-metal complexes with the plasma membrane[10]. In this study, we identified multiple proteins that were highly translated in response to the drug are involved in the alteration of the membrane composition, such as aminophospholipid translocase (*LmjF.34.3220*), lipase (*LmjF.31.2460*), lipase (*LmjF.13.0200*), several amastins located in the chromosome 8 and 34, surface antigen-like protein (*LmjF.09.0580*), and the GRAM domain-containing protein (*LmjF.04.0240*). Particularly, the translatome analysis showed that the amastin surface proteins are preferentially translated in response to drug challenge. This observation indicates that beyond its role in vertebrate host infection, amastins are important drug-resistance modulators. Though the literature supports the overexpression of amastins in antimony-resistant parasites, their role during drug modulation remains poorly investigated[41–44].

The gene variant analysis showed that most of the differentially translated transcripts do not have changes at the genomic level. However, genes with detected variants are associated with similar functional categories as those enriched by translatome analysis. These results suggest that the antimony-resistant phenotype is predominantly regulated by translational control.

Throughout this study, we utilized different strategies of analysis, such as gene ontology, protein clustering by homology, and co-expression network to identify putative translational modulators. Overall, we have shown that among several differences, resistant

**Fig. 9 | Model of the translational regulation as a central driver of *Leishmania*'s antimony-resistant phenotypes. a** Translational remodeling orchestrates a preemptive adaptation to drug challenges. **b** Drug-resistant parasites activate a highly selective translation in response to drug exposure.

parasites exhibit consistent changes in the expression of putative translational regulators, including the enrichment of protein kinases, helicases, mRNA binding proteins, translation initiation factors, and ribosomal proteins (Figs. 6, 7 and Supplementary Data 15). Interestingly, recent studies have shown that *Leishmania* parasites can use translational master regulators, such as the calcium-dependent kinase CDKP1 to modify the landscape of translating mRNAs modulating the resistance to paromomycin[25]. In addition, ribosomal proteins, such as eIF2alpha, are crucial for the translational reprogramming that occurs during parasite stage differentiation[71]. New studies in human cancer cells highlight the role of altering translational factors, such as eIF4E or eIF3, to induce dramatic translatomic changes that modulate drug-resistant phenotypes[72,73]. The activation of transporters and the antioxidant response demands a lot of energy in *Leishmania* parasites. The optimization of energy metabolism of resistant parasites observed during basal changes supports the idea of a preemptive adaptation.

Overall, our study demonstrates that during the development of drug resistance, resistant parasites undergo a dramatic translatome remodeling. After stepwise selection, resistant parasites exhibit a plethora of changes directed by translational regulators. Translational regulators themselves undergo changes occurring at both genomic and translational levels and influencing a complex remodeling. The fact that resistant parasites show dramatic changes in the translatome even in the absence of the drug challenge suggests that this remodeling could work as a parasite's strategy to be prepared for drug exposure. Therefore, we hypothesize that these changes are necessary to compensate for the loss of biological fitness of the parasites exposed to the drug for both immediate survival and as a preemptive

adaptation to the drug challenge (Fig. 9a). Our recent studies uncovered dramatic lipidome and metabolome remodeling even in the absence of antimony drug in the same drug-resistant *Leishmania tropica* parasites supporting the idea that those changes could be essential preemptive adaptations[10,13]. Our current findings suggest that translatome reprogramming could direct changes in lipidome and metabolome in resistant parasites. These coordinated changes preemptively prepare parasites to counteract the antimony drug very efficiently. Resistant parasites growing under drug challenge show targeted changes in the translational efficiency of only a few specific modulators of drug resistance (Fig. 9b). As a result, 33 transcripts downregulate their translation, and a specific population of 156 transcripts shifts towards a highly efficient translation by forming heavy polysomes. The produced proteins act in interconnected biological processes that modulate the effective response to the drug. An optimized energy metabolism supplies the energy required to fuel the antioxidant response via thiol metabolism; in turn, thiols further inactivate the drug by the thiol-metal complex conformation; ABC transporters and the traffic by vesicles induce the drug efflux. Drug efflux via exocytosis can be favored by the remodeling of membrane and surface protein. When transcriptional control is limited, translational remodeling offers an energy-saving alternative to efficiently respond to the drugs. Thus, highly selective translation activated by the drug leads to a coordinated response to combat the drug that involves interconnected biological processes.

Deciphering molecular mechanisms of selective mRNA translation to combat stress induced by antimony holds a great promise in the development of innovative strategies for the treatment of

leishmaniasis based on targeting master regulators of drug-induced selective translation and is a matter of future investigation.

# Methods

## Reagents

Schneider insect medium (Sigma-Aldrich, #S0146), fetal bovine serum or FBS (Sigma-Aldrich, #F0926), Dulbecco's Phosphate Buffered Saline (Sigma-Aldrich, #D8537), potassium antimony (III) tartrate trihydrate (Sigma-Aldrich, #230057), penicillin/streptomycin (Sigma-Aldrich, #P0781), MTT or 3-[4,5-dimethylthiazol-2-yl]-dephenyltetrazolium bromide (Sigma-Aldrich, #M2128), cycloheximide (Sigma-Aldrich, C7698), TRIzol™ LS Reagent (Invitrogen, #10296028), Power SYBR Green PCR Master Mix (Applied Biosystems, #4367659), Direct-zolTM RNA MiniPrep (Zymo Research, #R2052); poly(A) tailing Kit (Thermo Fisher Scientific, AM1350), HEPES (Fisher Scientific, BP310), Dithiotreitol or DTT (Fisher Scientific, BP172), Nonidet P 40 or NP-40 (Thermo Fisher Scientific, #J610055), potassium Chloride (KCl) (Sigma-Aldrich, P9541), magnesium chloride hexahydrate (Acros Organics, #413415000), recombinant Rnasin Ribonuclease Inhibitor (Promega, #N251B), cOmplete Mini EDTA-free protease inhibitors (Roche Diagnostics, #11873580001).

## *Leishmania* parasites culture

*Leishmania tropica* promastigotes were grown at a density of $5 \times 10^5$ cells/mL, in Schneider Insect medium, supplemented with 10% Fetal Bovine Serum (FBS), 10 units/mL Penicillin and 0.1 mg/mL streptomycin.

## Antimony resistance stepwise selection

The trivalent antimonial ($Sb^{III}$), considered to be the active form of the drug, was administered as potassium antimony (III) tartrate trihydrate[74]. The stepwise $Sb^{III}$ resistance selection was started at 10 μg/mL of $Sb^{III}$, equivalent to the EC50 estimated for the wild-type strain. The derived parasites were sequentially treated with 10, 20, 30, 40, 50, and 100 μg/mL of $Sb^{III}$. The MTT cell viability assay was used to estimate the response to the drug at three different points in the kinetics of resistance development: initial point ($Sb^{III}$-sensitive strain or WT), middle point i.e., parasites derived from treatment with 50 μg/mL of $Sb^{III}$ ($Sb^{III}$-moderately resistant strain or MR), and final point i.e., parasites derived from treatment with 100 μg/mL of $Sb^{III}$ ($Sb^{III}$-highly resistant strain or HR)[75]. In addition, response to the drug was measured in a similar manner in a highly resistant strain after culturing HR parasites for 25 passages in the absence of the drug. EC50 values were estimated using Probit regression or three-parameter dose-response model. CellTrace™ CFSE (Thermo Fisher Scientific) was used in cell proliferation assay following manufacture instructions. To estimate differences in parasite growth curves after drug-resistance selection, we used a Neubauer chamber to count the parasite density for 8 days post-seeded in 5 ml of Schneider Insect medium using T25 flasks.

## Infectivity assay

The infectivity assay with *Leishmania* parasites *was based* on Palacios et al., 2017 protocol[76]. The U937 cell line exhibiting monocyte morphology (ATCC: CRL-1593.2™) was stimulated with phorbol 12-myristate 13-acetate (Sigma-Aldrich®, # P8139) and seeded at $3 \times 10^5$ cells per well in 24 well plates (Corning®, # 3514). The cells were incubated at 37 °C and 5% CO2 for 72 h to promote differentiation to phagocytic cells. For in vitro infection, Leishmania promastigotes were cultured in Schneider's Insect Medium and harvested after 5 days of culture (stationary phase). The infection was initiated by mixing parasites and phagocytic cells in a ratio 1:15 and resuspended in RPMI 1640 medium (Gibco, # 11875093) without FBS. The infection was completed at 34 °C and 5% CO2 for 3 h. Next, the cells were washed with PBS to remove free-moving parasites. Infected cells were resuspended in RPMI supplemented with 10% FBS and incubated at 37 °C for

24 h. Finally, the cells were washed with PBS, fixed with 100% methanol, and stained with Giemsa following the commercial instructions (Sigma-Aldrich, # 64571-95). The number of infected cells was determined by light microscopy using ×1000 total magnification. A total of 100 cells were counted per condition. The experiments were completed in four biological replicates with two technical replicates.

## Polysome profiling

Three experimental conditions were analyzed, (1) the WT strain growing without drug treatment, (2) the HR strain growing without drug treatment, and (3) the HR strain treated with 100 μg/mL of $Sb^{III}$ for 12 h after 36 h of culture to induce the mechanisms necessary for drug resistance. In each case, the promastigotes were seeded at a density of $5 \times 10^5$ promastigotes/mL in a final volume of 60 mL of Schneider´s Insect medium supplemented with FBS.

Parasite's lysate preparation was based on our previous studies[77]. After 48 h of growth at 26 °C (mid-log phase), the cultures were treated with cycloheximide at a final concentration of 100 μg/mL for 10 min at 26 °C to arrest the ribosomes on translated mRNAs. The number of parasites was estimated by hemocytometer using 3.5% formaldehyde solution (Electron Microscopy Sciences, #15710) for cell fixing. Based on preliminary experiments, an optimal parasite concentration in terms of the polysome profiling quality ranged from 1.5 to $2 \times 10^7$ promastigotes/mL in 60 mL of medium.

After cell counting, a total of $7.5 \times 10^8$ promastigotes were harvested per experimental condition by centrifugation at 1.800×*g* and 4 °C for 8 min and washed in 30 mL of DBPS containing 100 μg/mL cycloheximide. The cells were centrifuged at 1.800×*g* and 4 °C for 5 min and resuspended in 1 mL of lysis buffer (20 mM HEPES-KOH (pH 7.5), 10 mM MgCl₂, 100 mM KCl, 2 mM DTT, 1% NP-40, 1× protease inhibitor cocktail (EDTA-free), 80 units/ml RNasin, and 100 μg/ml of cycloheximide). Details of cell lysate preparation are described previously[77]. The lysate was centrifuged at 11,200×*g* and 4 °C for 10 min and the supernatant was kept on ice. The supernatant was normalized based on the absorbance at 260 nm prior to sucrose-gradient ultracentrifugation.

To separate polysomes, ~500 μl of the clarified lysates was loaded on the top of 10–50% sucrose gradient containing 20 mM HEPES-KOH (pH 7.5), 100 mM KCl, 10 mM MgCl₂, 1 mM DTT, 80 units/ml RNasin, and subjected to ultracentrifugation using SW41 rotor for 2 h at 260,000×*g* and 4 °C. After centrifugation, ~500 μl of fractions were collected using a Piston Gradient Fractionator (BioComp Instruments). The polysome profiles were monitored by the absorbance at 260 and 280 nm using the Triax™ Flow Cell 1.56 A software. The absorbance at 260 nm Vs. fractions number plot was generated to identify the ribosome distribution. Trizol LS was added to fractions immediately and samples were kept at −80 °C until RNA extraction[77]. The fractions covering the peaks associated with monosomes (one ribosome), light polysomes (2–4 ribosomes), and heavy polysomes (5–9 ribosomes) were analyzed and pooled per each independent experiment.

## RNA isolation

Monosome, light, and heavy polysome pools were used for RNA isolation by Direct-zolTM RNA MiniPrep following the manufacturer's instructions. For RT-qPCR analysis, synthetic and polyadenylated outer membrane protein A (OmpA) mRNA was spiked into all samples before RNA purification and used for normalization as previously described[77]. RNA was recovered in 50 μL nuclease-free water, quantified by Qubit 3.0 (Life Technologies), and compared to reference samples. A total of 2.5 ng of polyadenylated OmpA were added per 1 μg of total RNA before sequencing.

## High-throughput sequencing

A total of 36 samples were prepared for sequencing. Sample quality was assessed by Agilent RNA 6000 Nano Reagent on Agilent 2100

Bioanalyzer (Agilent Technologies) and quantified by Qubit RNA HS assay (Thermo Fisher Scientific, #Q32857). Enrichment of poly-A RNA was performed by NEB Next® Poly(A) mRNA Magnetic Isolation Module (New England BioLabs Inc., #7490). Subsequently, library preparation was performed with 2 µg poly-A RNA by NEB Next Ultra II RNA library prep non-directional (New England BioLabs Inc.) following manufacturer instructions. Paired RNA-seq was carried out in an Illumina Hiseq 2000 platform using HCS2.2.68 software, with read length of 150 nucleotides, and an estimated of 40 M PE reads per sample (20 M in each direction).

## Bioinformatics analysis of translatome

All bioinformatic analyses were completed using the Galaxy platform (https://usegalaxy.org/)[78]. The paired reads were initially evaluated by the FastQC v0.73 (Babraham Bioinformatics) to verify the quality and the presence of adapters[79]. Then, the adapters were removed and the reads were filtered (Phred quality score >30, read length ≥50 nucleotides) using Trim Galore wrapper script (Version 0.6.3)[80]. After the processing of raw reads, the read quality was confirmed and summarized using the MultiQC algorithm v1.11 (Babraham Bioinformatics)[81]. Then, the paired reads were mapped to the *L. major* Friedlin strain genome (available in tritrypdb.org/ and released on 2019-11-04) using BWA-MEM (Version 0.7.17.1) mapper[82], the derived.bam files were submitted to coordinated sort using the function SortSAM of Picard tools kit (Version 2.18.2.1)[83]. MarkDuplicates of the Picard tool kit (2.18.2.2) were used for PCR duplicate removal. The counting matrices (raw number of reads mapped per transcript) were generated using the feature Counts algorithm[84]. The counting matrices were normalized using Deseq2 v1.34.0 normalization for differential expression analysis[33]. To detect statistically differentially expressed genes per comparison, a fold change cutoff of ≥1.5 (change in gene expression greater than 50%) and a Benjamini−Hochberg adjusted *P* value <0.05 were fixed as threshold. DTTs with statistically significant changes were filtered by two parameters: the significance or *P* value score estimated by a Wald test and corrected by the Benjamini−Hochberg method, and the fold change score estimated by the normalized read counts ratio.

## Validation of DTTs by RT-qPCR

cDNA samples were prepared using High-Capacity cDNA Reverse Transcription Kit (Applied Biosystems, #4368813). Reverse transcription-quantitative polymerase chain reactions (RT-qPCR) were done on Quant Studio 12 K Flex Real-Time PCR System using Power SYBR Green PCR Master Mix (Applied Biosystems, #4367659) according to the manufacturer's protocol. The differential gene expression was estimated using the ΔΔCT method[85]. For this purpose, mRNA for the hypothetical protein *Lmj*F.35.1945/LTRL590_350024600.1 (forward primer: GCGAAGCTGAGGCGGGAGAACGAG; reverse primer: AGCTTCTCCGCATCCGCAGCGAG) was used as the internal control. In addition, 13 additional mRNAs were analyzed by RT-qPCR (Supplementary Data 9).

## Validation of DTTs by proteomic analysis

Promastigotes were first mixed with extraction buffer (50 mM ammonium bicarbonate (Fisher Scientific) buffer (ABC), 5% sodium deoxycholate (SDC), (Fisher Scientific). The samples were homogenized by adding 3-mm zirconium beads using a beads beater. The homogenization was done at 4 °C at 4000 rpm for 30 s, followed by a 30-s pause. This step was repeated four times. After centrifuging for 15 min, the supernatant containing extracted proteins was collected and diluted ten times by adding 50 mM ABC buffer. Protein concentration was determined by BCA protein assay kit (ThermoScientific Fisher, #23227) following the manufacturer's instructions.

A 20-µg aliquot of extracted proteins from each sample was then subjected to reduction, alkylation, and tryptic digestion. ABC (50 mM)-

SDC (0.5%) solution was first added to samples to maintain a volume of 50 µL. Proteins were thermally denatured at 80 °C for 10 min. The reduction of proteins was performed by adding a 1.25-µL aliquot of 200 mM dithiothreitol solution and incubating at 60 °C for 45 min. The reduced proteins were then alkylated by adding a 5-µL aliquot of 200 mM iodoacetamide solution and incubation at 37 °C in the dark for 45 min. To quench the excessive IAA, a 1.25-µL aliquot of DTT solution was added again and samples were incubated at 37 °C for 30 min. Following the reduction and alkylation of proteins, trypsin (Promega) was added at a ratio of 1:25 (enzyme: proteins, w/w) into samples and incubated at 37 °C for 18 h. After incubation, formic acid was added at a final concentration of 0.5% (v/v) for both quenching the enzymatic reaction and removing the SDC detergent. Samples were then mixed thoroughly and centrifuged at 21,100×*g* for 10 min. The supernatant was collected, speed-vac dried, and resuspended in an aqueous solution containing 2% acetonitrile (ACN) and 0.1% formic acid (FA) before LC-MS/MS analysis using ThermoScientific XCalibur 4.2 at Proteomics/Metabolomics core facility, Texas Tech University.

The raw data files obtained from LC-MS/MS analysis were processed with the Proteome Discoverer software version 2.2 (Thermo Fisher Scientific). Database search was performed against UniProtKB/Swiss-Prot *L. major* database. The search included cysteine carbamidomethylation as a fixed modification and variable modifications, including methionine oxidation and acetylation of protein N-terminal. Trypsin was specified as the proteolysis enzyme and a maximum of two missed cleavages were allowed. For identification, the mass tolerance of 10 ppm and ±0.02 Da was chosen for MS and MS/MS analysis, respectively. The false discovery rate (FDR) was set to be 0.01 at both peptide and proteins and only scans with ≤0.01 FDR as determined by Percolator were used for protein identification. Additionally, only proteins with at least two identified peptides were considered for further analysis. Detailed workflow is described in Supplementary Fig. 6.

## Analysis of the differentially expressed genes detected at the translatome versus transcriptome level

After DESeq2 analysis, two groups of samples were used to compare the changes in the translatome Vs. transcriptome. The DTTs detected by analyzing heavy polysome fractions were used to indicate changes in highly translated transcripts. The DEG detected in the total mRNA (total RNA before polysome profiling fractionation) was considered to identify changes in the total transcriptome. In both cases, the statistically significant changes were identified by using a cutoff of absolute fold change ≥1.5 and a Benjamini−Hochberg adjusted *P* value ≤0.05. The resulting differentially translated/expressed genes were matched by gene ID and these genes were classified into 4 groups. Group 1: genes that were detected as differentially expressed only in heavy polysomes (DTT). Group 2: genes that were detected as differentially expressed only in the total transcriptome (transcriptionally regulated or mRNA stability). Group 3: genes that were detected as differentially expressed in both total transcriptome and heavy polysomes (mRNA abundance). Group 4: the remaining genes that did not show significant changes in either analysis. The analysis was performed using three biological replicates and considered both resistant parasites growing without and with drug challenges.

## Functional enrichment analysis

Gene ontology analysis was performed using the TriTrypDB web browser[86]. Networks based on protein homology were based on the interactome published in ref. 87. Networks based on protein co-expression were built in (https://string-db.org/). All networks were visualized on Cytoscape 3.7.2.

## Bioinformatics approach for call variants analysis

The variants were identified utilizing FreeBayes algorithm (Version 1.3.1), which has been successfully used in *Leishmania* that commonly

exhibits chromosome and gene copy number variations[34,88–90]. VCFfilter (Version 1.0) was employed to filter FreeBayes VCF for strand bias (Strand balance probability for the reference allele or SRP >30 and Strand balance probability for the reference allele or SAP > 30), placement bias (End Placement Probability or EPP > 30), variant quality (QUAL > 30), and depth of coverage (Total read depth at the locus or DP >100)[91]. The.vcf outputs from VCFfilter were annotated with SnpEff eff function using a custom database built with the.GFF dataset using the SnpEff v4.3 build function[92]. The identified variants were visually and manually inspected for accuracy using the Integrative Genomics Viewer (IGV). The variants exclusively detected in HR strain were considered for downstream analysis. Detailed workflow is described in Supplementary Fig. 6. The variants were summarized in a circus plot using the shinyCircos v1 function[93].

### Statistical analyses

Statistical analyses summarized in bar and box plots were performed using GraphPad Prizm software v9.1.2. Statistical analyses for RNA-Seq and proteomic data were performed using R v4.1 and proteome Discoverer software version 2.2, respectively.

### Reporting summary

Further information on research design is available in the Nature Portfolio Reporting Summary linked to this article.

## Data availability

The deep RNA-Seq datasets supporting the conclusions of this article are available in the Gene Expression Omnibus database from the National Center for Biotechnology under accession number GSE173848. The proteomics data supporting the findings of this study are available through Mass Spectrometry Interactive Virtual Environment under accession number MSV000089617 (*MassIVE*, https://massive.ucsd.edu/ProteoSAFe/static/massive.jsp?redirect=auth). The project data are grouped and available in the BioStudies database under accession number S-BSST850 (https://www.ebi.ac.uk/biostudies/). Source data are provided with this paper.

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

## Acknowledgements

We thank all members of the lab for helpful discussion and support, especially to Sarah Hernandez for the critical reading of the manuscript. We are thankful to Admera Health for deep RNA-Seq and core facilities at Texas Tech University for proteomics services. We acknowledge to the Galaxy Project (https://usegalaxy.org) for the web resource, computational server, and ToolShed repository used in this study for bioinformatic analyses. This study was supported by funds from "Minciencias and Gobernación del Tolima-Colombia as part of the program 755–2016" to S.A.GG and C.E.M; the Start-up fund from Texas Tech University Health Sciences Center to A.L.K.; and in part by the National Institute Of General Medical Sciences of the National Institutes of Health under Award Number R15GM146171 to Z.N.K. The content is solely the responsibility of the authors and does not necessarily represent the official views of the National Institutes of Health.

## Author contributions

Z.N.K., C.M., and S.A.G.G. designed experiments and analyzed the results. S.A.G.G., Z.N.K., E.B.T., and A.L.K. performed experiments. S.A.G.G. performed bioinformatics analysis. S.A.G.G. and Z.N.K. wrote the manuscript. All the authors reviewed the final manuscript version.

## Competing interests

The authors declare no competing interests.
