## [Peer Review File · Nature Communications]

Reviewer comments, first round

Reviewer #1 (Remarks to the Author):

In this study, Gutierrez Guarnizo et al investigated the contribution of translational control to the development of antimonial-resistance in *Leishmania tropica* using polysome profiling quantified by RNA-seq. Analysis of the translome was complemented with a proteomics approach and RT-qPCR experiments for selected targets. Gene ontology and protein-protein interaction analyses were performed with to identify cellular process potentially regulated through translational control in resistant strains. In parallel, gene variant calling analyses were carried out to resolve changes in the translome from changes in the genome triggered in response to drug pressure. The authors conclude that reprogramming of mRNA translation contributes to stress responses in antimonial-resistant strains (e.g. metabolism, antioxidant response, mRNA translation, drug efflux, etc.). This study has the potential to provide insight into the molecular mechanisms that confer chemoresistance in *Leishmania* parasites; however, it is not clear how the authors were able to resolve changes in mRNA translation efficiency from changes in mRNA abundance owing to other mechanisms of post-transcriptional regulation such as mRNA turnover (i.e. mRNA stability versus mRNA decay). Typically, to identify changes in mRNA translation, in parallel to analyzing the translome (i.e. changes in mRNA levels in monosomal vs polysomal fractions), the authors should carry out a transcriptomic analysis of total mRNA (input mRNA prior to fractionation). Then, an algorithm such Anota2seq (PMID: 35119670) should be employed to distinguish bona fide changes in mRNA translation efficiency from those that are simply derived from differences in mRNA abundance, which in the case of *Leishmania* mainly stem from changes in mRNA stability. Can the authors please clarify if they obtained transcriptomics data in parallel to the analyses presented in the manuscript? If yes, please reanalyze the data using an adequate methodology.

Reviewer #2 (Remarks to the Author):

Translational Reprogramming as a Driver of Antimony-drug Resistance ^[1]_{SEP}

The authors embark on a novel approach to identify post-transcriptional gene regulation in the context of metalloids, i.e. antimony, resistance using polysome profiling analysis. They raised moderately and highly resistant populations of *Leishmania tropica* promastigotes in vitro, by exposure to stepwise increases of trivalent antimony, a common strategy to induce drug tolerance in vitro. They compared the monoribosome and polyribosome association of RNA species in unselected, untreated promastigotes, selected promastigotes and selected + Sb(III)-treated promastigotes and found the most differences in RNA-ribosome association between unselected and selected, non-treated cells.

The authors did perform both RT-qPCR on selected mRNAs and RNA-seq analysis to distinguish between expression changes mediated by RNA abundance or translation efficiency and found that expression in many cases is regulated purely at the translation level. Translation activity was matched against protein abundance using mass spec analysis, confirming a strong translational regulation component in the parasite's adaption to metalloid stress.

The data provide strong evidence for a molecular preadaptation to antimony challenge at the level of protein synthesis, independent of gene copy number variations and RNA steady state levels. The manuscript is bound to draw more attention to regulated translation in trypanosomatid protozoa.

Reviewer #3 (Remarks to the Author):

Guarnizo et al., have used polysome profiling and RNA-seq to study antimony resistance in the

protozoan parasite *Leishmania*. This is a new and welcomed strategy for studying drug resistance in *Leishmania*. The polysome profiling/RNAseq work appears to have been done expertly. Yet I have comments on both the experimental design and interpretation that I would like to share with the authors.

1. I realize the amount of work in polysome profiling with the multiple RNA libraries and admittedly they have done a reasonable job in trying to fit some of their data with what is known in the literature. Some data, e.g. AQP1 overexpression, do not fit with our current understanding of antimony resistance. However, it is hard to reach firm conclusions when studying a single resistant line. In my opinion studying a minimum of three lines selected independently for resistance are needed to reach general conclusions. The current line may just be an exception. By removing the light polysome fraction and the +/- drug component (see below) this would have led to a workable number of RNA libraries.

2. Many information regarding their single *L. tropica* mutant was lacking. Was it derived from a clone? Does it grow similarly as their wild-type? Mutants often have growth defect that can be observed only by carefully calibrated growth curves over time. A crippled parasite is likely to differ in many features in comparison to a wild-type, especially after many repeated passages. Are the resistant parasites equally infective to macrophages? Is resistance stable or it decrease rapidly when grown in repeated passage in absence of drug? I think these points are important. Indeed, the authors make a strong case about change in the translome for fitness purposes but this has to be shown.

3. From the results provided I would suggest that their resistant line is still stressed. Flagellar size appears smaller in resistant parasites (Fig. S5) and these would not fare well in competition experiments with wild-type cells. The current dogma in the literature (derived mostly from the Dujardin group) would suggest that resistant parasites are fitter and this is why they remain in the field even in the absence of selection. I was surprised to see the amastins in their translome. Amastins are found in amastigotes and here they work with promastigotes. It is well known that stresses (e.g. pH, Temperature) can induce differentiation and the antimony stress can possibly partly induce some of those differentiation signals leading to amastin expression.

4. I am unclear with their concept of preemptive adaptation. They used a strain that was already resistant to the drug. Adding drug to it should not change much (and this is what they observed). In my opinion in the case of preemptive adaptation is that a wild-type cell would have all the tools to adapt rapidly to an insult. This is not the case, it takes time to generate a resistant line. In my opinion carrying work with a wild-type cell with sub-toxic concentration of SbIII would have been more telling than adding drug to a resistant line.

Reviewer #1 (Remarks to the Author):

In this study, Gutierrez Guarnizo et al investigated the contribution of translational control to the development of antimonial-resistance in Leishmania tropica using polysome profiling quantified by RNA-seq. Analysis of the translome was complemented with a proteomics approach and RT-qPCR experiments for selected targets. Gene ontology and protein-protein interaction analyses were performed with to identify cellular process potentially regulated through translational control in resistant strains. In parallel, gene variant calling analyses were carried out to resolve changes in the translome from changes in the genome triggered in response to drug pressure. The authors conclude that reprogramming of mRNA translation contributes to stress responses in antimonial-resistant strains (e.g. metabolism, antioxidant response, mRNA translation, drug efflux, etc.).

Comments from the reviewer:

This study has the potential to provide insight into the molecular mechanisms that confer chemoresistance in Leishmania parasites; however, it is not clear how the authors were able to resolve changes in mRNA translation efficiency from changes in mRNA abundance owing to other mechanisms of post-transcriptional regulation such as mRNA turnover (i.e. mRNA stability versus mRNA decay). Typically, to identify changes in mRNA translation, in parallel to analyzing the translome (i.e. changes in mRNA levels in monosomal vs polysomal fractions), the authors should carry out a transcriptomic analysis of total mRNA (input mRNA prior to fractionation). Then, an algorithm such Anotas2seq (PMID: 35119670) should be employed to distinguish bona fide changes in mRNA translation efficiency from those that are simply derived from differences in mRNA abundance, which in the case of Leishmania mainly stem from changes in mRNA stability. Can the authors please clarify if they obtained transcriptomics data in parallel to the analyses presented in the manuscript? If yes, please reanalyze the data using an adequate methodology.

Dear Reviewer,

Thank you very much for the critical reading of our manuscript and providing comments and suggestions. We completed a number of new experiments and edited the manuscript to answer the questions and addressed comments of the reviewers. We have submitted the revised version of the manuscript with tracking.

Point by point response:

Thank you for the suggestion to perform a transcriptomic analysis of total mRNA. We completely agree that analysis of the total mRNAs is important to distinguish changes in translomes from those derived from differences in mRNA abundance. We performed deep RNA-sequencing of total mRNAs previously collected as inputs for our original polysome profiling experiments. We used the same sequencing protocol and differential gene expression analysis for the total mRNAs as for our polysomal fraction mRNAs, as is already described in the manuscript, to make them as comparable as possible. Therefore, we applied the DESeq2 algorithm that we used previously for the data analysis. We considered to use the suggested Anota2seq algorithm, but decided to use the DESeq2 algorithm because all our data were analyzed using this algorithm. Our differential gene expression analysis based on the DESeq2 algorithm also showed a good correlation with our qPCR data, and we felt more confident in basing our additional analyses on this algorithm.

As a result, the strategy we chose was to compare the total transcriptome (total mRNA used as input for polysome profiling) and heavy polysome mRNA (more efficiently translated transcripts). After DESeq2 analysis and the respective matching, the genes were classified into 4 groups. Group 1: genes that were detected as differentially expressed only in heavy polysomes ("translation" in Anota2seq). Group 2: genes that were detected as differentially expressed only in the total transcriptome ("mRNA stability" or "transcription" in Anota2seq). Group 3: genes that were detected as differentially expressed in both total transcriptome and heavy polysomes ("mRNA abundance" in Anota2seq). Group 4: genes that were not considered as differentially expressed in any of the previously mentioned groups. We used the cutoffs previously used: absolute fold change ≥ 1.5 and false discovery rate ≤ 0.05). This analysis was completed for both "basal changes" and "drug challenge". The transcriptomic analysis under basal conditions identified only 62 differentially expressed transcripts, while analysis of heavy polysome fraction identified 2055 differentially translated transcripts. The new results demonstrate that the vast majority of changes in resistant parasites are observed in translome rather than transcriptome and strongly support our model that translational reprogramming plays a major role in drug resistance in *Leishmania* parasites.

Comparison of transcriptome with translome under the drug challenge also supports that translational control plays a predominant role in drug resistance.

As a result, we have modified the experimental design in Figure 2 to include a schematic of translome versus transcriptome data analysis (Figure 2 C). We also combined the original figures 3 and 4 (now figure 3 A-D) and incorporated the new transcriptome VS translome results

(now figure 3E-F). New raw and processed datasets were uploaded to GEO-Seq (GSE173848). The new supplementary files (now S4 and S7) were also included in the revised manuscript. Corresponding sections describing new data were added to the methods, results, and discussion.

Summary of the new experiments and figures added to the revised version of the manuscript:

1. Figure 2 was modified to include a new experimental design of the transcriptome versus translome (Figure 2C).

2. Deep RNA-seq of total mRNAs and transcriptomic analysis were performed. New data on comparative analysis of translome versus transcriptome are included now as Figure 3 E and F. Figure 3 also has translome data combined from original figures 3 and 4 (Figure 3 A-D now).

3. New Supplementary File S4 contains comparison of Translome Vs Transcriptome at Basal level (in the absence of drug). Figure 3 E is based on the Supplementary File S4 data.

4. New Supplementary File S7 contains comparison of Translome Vs Transcriptome under the drug challenge. Figure 3 F is based on the Supplementary File S7 data.

5. New data on comparison of proliferation of parental-sensitive and drug-resistant strains are included as Supplementary Figure 1.

6. New data on examination of stability of drug-resistance phenotype are included as Supplementary Figure 2.

7. New data on infectivity of drug-resistant strain are included as Supplementary Figure 3.

Original Supplementary figures 1, 2, 3, 4, and 5 became correspondingly supplementary figures 4, 5, 6, 8 and 7.

Reviewer #2 (Remarks to the Author):

Translational Reprogramming as a Driver of Antimony-drug Resistance

*The authors embark on a novel approach to identify post-transcriptional gene regulation in the context of metalloids, i.e. antimony, resistance using polysome profiling analysis. They raised moderately and highly resistant populations of *Leishmania tropica* promastigotes in vitro, by exposure to stepwise increases of trivalent antimony, a common strategy to induce drug tolerance in vitro. They compared the monoribosome and polyribosome association of RNA species in unselected, untreated promastigotes, selected promastigotes and selected + Sb(III)-treated*

promastigotes and found the most differences in RNA-ribosome association between unselected and selected, non-treated cells.

The authors did perform both RT-qPCR on selected mRNAs and RNA-seq analysis to distinguish between expression changes mediated by RNA abundance or translation efficiency and found that expression in many cases is regulated purely at the translation level. Translation activity was matched against protein abundance using mass spec analysis, confirming a strong translational regulation component in the parasite's adaption to metalloïd stress.

The data provide strong evidence for a molecular preadaptation to antimony challenge at the level of protein synthesis, independent of gene copy number variations and RNA steady state levels. The manuscript is bound to draw more attention to regulated translation in trypanosomatid protozoa.

Response to the reviewer #2:

Thank you very much for the critical reading of our manuscript and providing comments and suggestions.

Reviewer #3 (Remarks to the Author):

Guarnizo et al., have used polysome profiling and RNA-seq to study antimony resistance in the protozoan parasite Leishmania. This is a new and welcomed strategy for studying drug resistance in Leishmania. The polysome profiling/RNAseq work appears to have been done expertly. Yet I have comments on both the experimental design and interpretation that I would like to share with the authors.

Comments from the reviewer:

1. I realize the amount of work in polysome profiling with the multiple RNA libraries and admittedly they have done a reasonable job in trying to fit some of their data with what is known in the literature. Some data, e.g. AQP1 overexpression, do not fit with our current understanding of antimony resistance. However, it is hard to reach firm conclusions when studying a single resistant line. In my opinion studying a minimum of three lines selected independently for resistance are needed to reach general conclusions. The current line may just be an exception. By removing the light polysome fraction and the +/- drug component (see below) this would have led to a workable number of RNA libraries.

2. Many information regarding their single L. tropica mutant was lacking. Was it derived from a clone? Does it grow similarly as their wild-type? Mutants often have growth defect that can be observed only by carefully calibrated growth curves over time. A crippled parasite is likely to

differ in many features in comparison to a wild-type, especially after many repeated passages. Are the resistant parasites equally infective to macrophages? Is resistance stable or it decrease rapidly when grown in repeated passage in absence of drug? I think these points are important. Indeed, the authors make a strong case about change in the translome for fitness purposes but this has to be shown.

3. From the results provided I would suggest that their resistant line is still stressed. Flagellar size appears smaller in resistant parasites (Fig. S5) and these would not fare well in competition experiments with wild-type cells. The current dogma in the literature (derived mostly from the Dujardin group) would suggest that resistant parasites are fitter and this is why they remain in the field even in the absence of selection. I was surprised to see the amastins in their translome. Amastins are found in amastigotes and here they work with promastigotes. It is well known that stresses (e.g. pH, Temperature) can induce differentiation and the antimony stress can possibly partly induce some of those differentiation signals leading to amastin expression.

4. I am unclear with their concept of preemptive adaptation. They used a strain that was already resistant to the drug. Adding drug to it should not change much (and this is what they observed). In my opinion in the case of preemptive adaptation is that a wild-type cell would have all the tools to adapt rapidly to an insult. This is not the case, it takes time to generate a resistant line. In my opinion carrying work with a wild-type cell with sub-toxic concentration of SbIII would have been more telling than adding drug to a resistant line.

Response to the reviewer:

Thank you very much for the critical reading of our manuscript and providing comments and suggestions. We completed a number of new experiments and edited the manuscript to answer the questions and addressed comments of the reviewers. We have submitted the revised version of the manuscript with tracking.

Point by point response to the reviewer's comments:

1. I realize the amount of work in polysome profiling with the multiple RNA libraries and admittedly they have done a reasonable job in trying to fit some of their data with what is known in the literature. Some data, e.g. AQP1 overexpression, do not fit with our current understanding of antimony resistance. However, it is hard to reach firm conclusions when studying a single resistant line. In my opinion studying a minimum of three lines selected independently for resistance are needed to reach general conclusions. The current line may just be an exception. By removing the light polysome fraction and the +/- drug component (see below) this would have led to a workable number of RNA libraries.

The downregulation of AQP1 has been consistently observed in multiple antimony-resistant strains. However, several studies have shown that there are exceptions to this rule in both clinical isolates and parasites selected for antimony resistance. For instance, Mandal et al., 2010 collaborators found that some clinical isolates of *Leishmania donovani* resistant to antimony up-regulate the AQP1 by RT-qPCR (Mandal, Maharjan, Singh, Chatterjee, & Madhubala, 2010). Maharjan et al 2008 showed that in clinical isolates the downregulation of AQP1 is not consistently found indicating heterogeneity in antimony resistance mechanisms. Indeed, the authors report that while the sensitive strain (2001-S) showed AQP1 decrease, the resistant strain (GE1-R) showed increased AQP1 suggesting that different expression patterns can be observed in resistant strains (Maharjan, Singh, Chatterjee, & Madhubala, 2008). The same pattern has been observed for some strains selected for antimony resistance *in vitro*. For example, the *L. infantum* Sb2000.1 showed high expression of AQP1 both in amastigote and promastigote (Marquis, Gourbal, Rosen, Mukhopadhyay, & Ouellette, 2005). In our study, the AQP1 did not show significant changes in resistant parasites growing without drug challenge. Under drug challenges, the resistant strain did not show changes in the total transcriptome (P-value: 0.85), heavy polysome fraction (P-value: 0.36), or monosome (P-value: 0.98). However, light polysome showed significant changes (P-value: 0,013, Fold change: +1.9). Our findings agree with the previously discussed studies and support the idea that the association between the AQP1 downregulation and antimony resistance does not apply to all strains. We have decided to modify the main text to clarify that even though the typical AQP1 downregulation was not detected in our model, examples of AQP1 not showing downregulation have been previously reported in clinical isolates and *in vitro* generated strains with resistance to antimony. Overall, our findings are consistent with several well-documented antimony resistance markers including up-regulation of ABC-transporters and S-adenosylmethionine synthase and downregulation of kinetoplastid membrane protein 11.

Regarding using three cell lines derived independently to confirm our findings. While it is a good suggestion it will lead to a huge increase in the number of samples and delay our current work. On the other hand, elimination of light polysome samples would not allow us to see changes in translational dynamics. In our opinion, it is also very important to have +/- drug conditions for our drug resistant strain since it proves that dramatic translational changes we observe occur as a preemptive strategy of the parasite during development of drug resistance. Our major message in the current study is to deliver the importance of translational control in drug resistance in *Leishmania* parasites. However, it would be interesting to examine in the future studies if parasites always use the same route of translational reprogramming to develop drug resistance or if

different solutions are possible via translational regulation to achieve the same outcome. Therefore, even though we haven't included other resistant strains in the current study, however, we have incorporated a sentence clarifying that the use of only one resistant strain is a limitation of this study, and complementary studies are necessary for the future.

2. Many information regarding their single L. tropica mutant was lacking. Was it derived from a clone? Does it grow similarly as their wild-type? Mutants often have growth defect that can be observed only by carefully calibrated growth curves over time. A crippled parasite is likely to differ in many features in comparison to a wild-type, especially after many repeated passages. Are the resistant parasites equally infective to macrophages? Is resistance stable or it decrease rapidly when grown in repeated passage in absence of drug? I think these points are important. Indeed, the authors make a strong case about change in the translome for fitness purposes but this has to be shown.

Thank you for these great suggestions. To address the reviewer's suggestions/questions we performed several experiments and included the new results as supplementary figures 1-3.

We agree with the reviewer that using a single clone could be problematic since clones could have a heterogeneity. Therefore, all our studies in the manuscript were done with the population of cells of *L. tropica* instead a clone.

The detailed characterization of the resistant strain included examination of proliferation, rate of growth and infectivity of drug resistant strains compare to parental sensitive strain, as well as evaluation of stability of drug resistance phenotype. Our new experiments demonstrated that both middle and highly resistant strains (MR and HR) had the same rate of proliferation in the absence or presence of antimony (new supplementary Figure 1A) and it was comparable to the parental sensitive grown without drug. The only strain that displayed a reduced proliferation in the presence of the drug was parental sensitive as expected. Our new growth curves experiments showed that resistant parasites cultured under the drug pressure exhibited a very similar growth profile compared to untreated sensitive strain (new supplementary Figure 1B). These data argue that parasites have very similar growth characteristics and not under the stress. The new results are summarized in the Supplementary Figure 1. Consistently, the MTT viability assay showed a similar signal intensity of formazan for sensitive and resistant parasites seeded at the same density, indicating similar viability (see left wells in the panels of figure 1A).

A new infectivity assay was completed to test whether the HR strain show changes in the infection to a human phagocytic cell line *in vitro*. Wild type sensitive, middle resistant (MR) and highly resistant (HR) strains were used in the experiments. The results indicated that the derived resistant strains have a reduction in infectivity that correlates well with the level of resistance. We have included the supplementary figure S3 to summarize these new results. Since the reduction in the infectivity correlates with the level of resistance, it is possible that it represents parasite's trade-off adaptations to acquire drug resistance at expense of infectivity. Such examples are known in the literature. Some multi-drug resistant *M. tuberculosis* strains can be 10-fold less transmissible than susceptible strains (Borrell & Gagneux, 2009). Compensatory evolution has been shown to mitigate the fitness defects including change in infectivity. Low infectivity in *Plasmodium falciparum* gametocytes to *Anopheles gambiae* is connected to resistance to sulfadoxine-pyrimethamine (Beavogui et al., 2010). Consistently, various studies have previously reported the decreased infectivity in *Leishmania* resistant parasites as well (Gazola et al., 2001; Liarte & Murta, 2010; Silva, Camacho, Figarella, & Ponte-Sucré, 2004).

Additionally, to test whether the resistance was stable, we submitted the HR (highly resistant) strain to 25 consecutive passages in absence of the drug as shown in the new supplementary figure 2. Notably, after these passages in culture the HR strain still showed a significantly higher EC50 value when compared to the wild-type (WT: $9,34 \pm 2.35$ HR: 278 ± 80 $\mu\text{g/mL}$ of Sb^{III}). We noticed that even though the EC50 value for the resistant strain was half of the obtained in the first experiment (immediately after the *in vitro* selection drug resistance was completed), still the considered highly resistant strain showed about 30 times higher EC50 than the wild-type parental strain. In previous studies in our laboratory, we considered a strain as resistant when the EC50 increased 10 times compared to the WT (L. H. Patino, Muñoz, Cruz-Saavedra, Muskus, & Ramírez, 2020; Luz H. Patino, Muskus, & Ramírez, 2019).

3. From the results provided I would suggest that their resistant line is still stressed. Flagellar size appears smaller in resistant parasites (Fig. S5) and these would not fare well in competition experiments with wild-type cells. The current dogma in the literature (derived mostly from the Dujardin group) would suggest that resistant parasites are fitter and this is why they remain in the field even in the absence of selection.

We have been interested in exploring whether antimony-resistant parasites show physiological stress when compared to the wild-type strain. Our Sb^{III} -resistant parasites do not show significant changes in proliferation even in the presence of drug when compared to the parental-sensitive

strain (new supplementary Figure 1) supporting that they are not stressed and can remain in the field even in the absence of selection. In our previously published work, using the same strain, we used lipidomic and metabolomic analyses to evaluate physiological changes in the antimony-resistant parasites (Gutierrez Guarnizo, Karamysheva, Galeano, & Muskus, 2021; Sneider Alexander Gutierrez Guarnizo et al., 2021). Our findings indicate that resistant parasites show metabolic and lipidomic remodeling associated with optimized energy metabolism and a better ability to counteract oxidative stress. Notably, our polysome profiling study also showed that several components of the antioxidant response and energy metabolism are highly translated in response to the drug while multiple flagellar components (Dynein, flagellar rod, kinesin proteins, etc.) demonstrated a significant decrease in heavy polysome (Figure 7E). Since flagellum movement is ATP-dependent and antioxidant response is highly energy demanding, the smaller flagellum in resistant parasites could be a parasite's trade-off strategy to optimize the energy required to combat the drug.

4. I was surprised to see the amastins in their translatoe. Amastins are found in amastigotes and here they work with promastigotes. It is well known that stresses (e.g. pH, Temperature) can induce differentiation and the antimony stress can possibly partly induce some of those differentiation signals leading to amastin expression.

Concerning the amastin expression in promastigote parasites, it is true that the amastins were first associated with amastigotes, but there is a consistent evidence indicating that these surface proteins are also expressed in promastigotes. Several studies have also reported the upregulation of amastins in Sb^{III}-resistant parasites (Downing et al., 2011; Luz H. Patino, Imamura, et al., 2019; Rastrojo et al., 2018; Verma et al., 2017). Predominantly these studies analyze the expression profile in promastigotes of antimony-resistant strains, and the amastin expression is detected in both sensitive and resistant strains (Luz H. Patino, Imamura, et al., 2019; Rastrojo et al., 2018; Verma et al., 2017). Complementary to the previous studies, here we provide an evidence that the amastins are also highly translated after the drug challenge. While we cannot completely rule out that increased amastin expression can be caused by partial differentiation, it seems unlikely since parasites do not show changes in proliferation in the presence of drug compare to the parental strain. However, it is possible that another role for amastins as surface proteins is to protect parasites against the drug by some currently unknown mechanism. In general, the amastin role in other biological processes besides virulence and infection remains largely unexplored and requires further investigation.

5. I am unclear with their concept of preemptive adaptation. They used a strain that was already resistant to the drug. Adding drug to it should not change much (and this is what they observed). In my opinion in the case of preemptive adaptation is that a wild-type cell would have all the tools to adapt rapidly to an insult. This is not the case, it takes time to generate a resistant line. In my opinion carrying work with a wild-type cell with sub-toxic concentration of SbIII would have been more telling than adding drug to a resistant line.

We introduced the preemptive adaptation term to define those genotypic/phenotypic changes that occur during the development of drug resistance. These changes are necessary for a rapid drug-specific response once the parasites are exposed to it. Based on our transcriptome analysis, wild type cells do not have all tools to adapt rapidly. Otherwise, wild type cells would survive the insult and highly resistant cells would not be selected. Instead, cells underwent a dramatic translational reprogramming to create the tools, adapt and survive. Preemptive adaptation is the process of creating these tools. In another words, HR cells already have the tools, and this allows parasites to be ready for the drug challenge. In addition, smaller changes in response of HR cells to drug support the conclusion that cells already have tools to survive the drug exposure. Furthermore, these detected changes could reflect immediate and short-time lived strategies that cells used to combat the drug. In other words, even though the response triggered to combat the drug involves the shift toward a highly efficient translation of a specific set of transcripts, the plethora of changes that precede the drug challenge could be essential to coordinate the resistant phenotypes.

For instance, Dr. Dujardin's studies indicated that antimony parasites show a higher phosphatidylcholine (PC) degradation. Consistently, our previous findings showed that in resistant parasites the PC is increased before the drug challenge but reduced upon drug exposure (Sneider Alexander Gutierrez Guarnizo et al., 2021). PC is an important component of the membrane but also a reservoir of methyl donor groups that can be used to feed the antioxidant response, a process that interconnects the Kennedy and the trypanothione pathway (Moitra, Pawlowic, Hsu, & Zhang, 2019). Notably, our polysome profiling showed that at the basal level, several components of the PC metabolism were highly translated in resistant parasites before drug exposure but not exacerbated during drug challenges. These genes include the plasma-membrane choline transporter (LmjF.36.2210), choline/ethanolamine kinase (LmjF.35.1470) phosphatidylcholine (PC), cholinephosphate cytidyltransferase A (LmjF.18.1330), cholinephosphate cytidyltransferase (LmjF.18.1330), and choline dehydrogenase

(LmjF.21.1563). In this case, the efficient translation of PC-pathway components could be considered as a preemptive adaptation that allows for better responses to the drug when it is applied. Discriminating a preemptive from a drug-inducible changes are important to understand how the drug resistance phenotypes are coordinated. Notably, particular examples of preemptive adaptation need to be experimentally validated.

Regarding carrying out work with parental wild-type cells using subtoxic concentration of antimony. Parental cells did not acquire the pre-adaptive changes as highly resistant strain did and therefore, we expect that the response of sensitive strain to subtoxic drug concentration will be different from its resistant counterpart. Our recent lipidomic studies support this conclusion – we have found that changes in lipid profile under subtoxic concentrations are quite distinct in parental sensitive and highly resistant strains (S. A. Gutierrez Guarnizo et al., 2021). Therefore, in order to reveal the real changes supporting the resistance it is more valuable to examine the responses to the high drug concentration using the strain that already developed the drug resistance. The use of subtoxic concentration with the resistant strain capable to withstand a very high drug concentration may lead to a limited blunt response and not reveal all changes. Thus, our current work is focused on the changes in translome in drug resistant strain. However, investigation of sensitive strain responses to subtoxic antimony concentration and comparison with resistant strain responses could be a subject of further separate study.

Summary of the new experiments and figures added to the revised version of the manuscript:

1. Figure 2 has been modified to include a new experimental design of the transcriptome versus translome (Figure 2C).

2. Deep RNA-seq of total mRNAs and transcriptomic analysis have been performed. New data on comparative analysis of translome versus transcriptome are included now as Figure 3 E and F. Figure 3 also has translome data combined from original figures 3 and 4 (Figure 3 A-D now).

3. New Supplementary File S4 contains comparison of Translome Vs Transcriptome at Basal level (in the absence of drug). Figure 3 E is based on the Supplementary File S4 data.

4. New Supplementary File S7 contains comparison of Translome Vs Transcriptome under the drug challenge. Figure 3 F is based on the Supplementary File S7 data.

5. New data on comparison of proliferation of parental-sensitive and drug-resistant strains are included as Supplementary Figure 1.

6. New data on examination of stability of drug-resistance phenotype are included as Supplementary Figure 2.

7. New data on infectivity of drug-resistant strain are included as Supplementary Figure 3.

Original Supplementary figures 1, 2, 3, 4, and 5 became correspondingly supplementary figures 4, 5, 6, 8 and 7.

This study provides for the first time how important is translational control in the development of drug resistance in *Leishmania* parasites. I hope that reviewer is satisfied with the new experiments and revised version of the manuscript. Thank you for the reviewing the manuscript and for suggestions.

The references mentioned in our responses to the reviewer:

- Beavogui, A. H., Djimde, A. A., Gregson, A., Toure, A. M., Dao, A., Coulibaly, B., . . . Doumbo, O. K. (2010). Low infectivity of *Plasmodium falciparum* gametocytes to *Anopheles gambiae* following treatment with sulfadoxine-pyrimethamine in Mali. *Int J Parasitol*, *40*(10), 1213-1220. doi:10.1016/j.ijpara.2010.04.010
- Borrell, S., & Gagneux, S. (2009). Infectiousness, reproductive fitness and evolution of drug-resistant *Mycobacterium tuberculosis*. *Int J Tuberc Lung Dis*, *13*(12), 1456-1466. Retrieved from <https://www.ncbi.nlm.nih.gov/pubmed/19919762>
- Downing, T., Imamura, H., Decuyper, S., Clark, T. G., Coombs, G. H., Cotton, J. A., . . . Berriman, M. (2011). Whole genome sequencing of multiple *Leishmania donovani* clinical isolates provides insights into population structure and mechanisms of drug resistance. *Genome research*, *21*(12), 2143-2156. doi:10.1101/gr.123430.111
- Gazola, K. C., Ferreira, A. V., Anacleto, C., Michalick, M. S., Andrade, A. F., & Moreira, E. S. (2001). Cell surface carbohydrates and in vivo infectivity of glucantime-sensitive and resistant *Leishmania* (*Viannia*) *guyjanensis* cell lines. *Parasitol Res*, *87*(11), 935-940. doi:10.1007/s004360100475
- Gutierrez Guarnizo, S. A., Karamysheva, Z. N., Galeano, E., & Muskus, C. E. (2021). Metabolite Biomarkers of *Leishmania* Antimony Resistance. *Cells*, *10*(5). doi:10.3390/cells10051063
- Gutierrez Guarnizo, S. A., Tikhonova, E. B., Zabet-Moghaddam, M., Zhang, K., Muskus, C., Karamyshev, A. L., & Karamysheva, Z. N. (2021). Drug-Induced Lipid Remodeling in *Leishmania* Parasites. *9*(4), 790. Retrieved from <https://www.mdpi.com/2076-2607/9/4/790>
- Gutierrez Guarnizo, S. A., Tikhonova, E. B., Zabet-Moghaddam, M., Zhang, K., Muskus, C., Karamyshev, A. L., & Karamysheva, Z. N. (2021). Drug-Induced Lipid Remodeling in *Leishmania* Parasites. *Microorganisms*, *9*(4).
- Liarte, D. B., & Murta, S. M. (2010). Selection and phenotype characterization of potassium antimony tartrate-resistant populations of four New World *Leishmania* species. *Parasitol Res*, *107*(1), 205-212. doi:10.1007/s00436-010-1852-8
- Maharjan, M., Singh, S., Chatterjee, M., & Madhubala, R. (2008). Role of aquaglyceroporin (AQP1) gene and drug uptake in antimony-resistant clinical isolates of *Leishmania donovani*. *Am J Trop Med Hyg*, *79*(1), 69-75.
- Mandal, S., Maharjan, M., Singh, S., Chatterjee, M., & Madhubala, R. (2010). Assessing aquaglyceroporin gene status and expression profile in antimony-susceptible and -resistant clinical isolates of

- Leishmania donovani from India. *Journal of Antimicrobial Chemotherapy*, 65(3), 496-507.
doi:10.1093/jac/dkp468
- Marquis, N., Gourbal, B., Rosen, B. P., Mukhopadhyay, R., & Ouellette, M. (2005). Modulation in aquaglyceroporin AQP1 gene transcript levels in drug-resistant Leishmania. *Mol Microbiol*, 57(6), 1690-1699. doi:10.1111/j.1365-2958.2005.04782.x
- Moitra, S., Pawlowic, M. C., Hsu, F. F., & Zhang, K. (2019). Phosphatidylcholine synthesis through cholinephosphate cytidyltransferase is dispensable in Leishmania major. *Sci Rep*, 9(1), 7602. doi:10.1038/s41598-019-44086-6
- Patino, L. H., Imamura, H., Cruz-Saavedra, L., Pavia, P., Muskus, C., Méndez, C., . . . Ramírez, J. D. (2019). Major changes in chromosomal copy number, gene expression and gene dosage driven by SbIII in Leishmania braziliensis and Leishmania panamensis. *Scientific Reports*, 9(1), 9485. doi:10.1038/s41598-019-45538-9
- Patino, L. H., Muñoz, M., Cruz-Saavedra, L., Muskus, C., & Ramírez, J. D. (2020). Genomic Diversification, Structural Plasticity, and Hybridization in Leishmania (Viannia) braziliensis. *Frontiers in cellular and infection microbiology*, 10, 582192. doi:10.3389/fcimb.2020.582192
- Patino, L. H., Muskus, C., & Ramírez, J. D. (2019). Transcriptional responses of Leishmania (Leishmania) amazonensis in the presence of trivalent sodium stibogluconate. *Parasites & Vectors*, 12(1), 348. doi:10.1186/s13071-019-3603-8
- Rastrojo, A., Garcia-Hernandez, R., Vargas, P., Camacho, E., Corvo, L., Imamura, H., . . . Requena, J. M. (2018). Genomic and transcriptomic alterations in Leishmania donovani lines experimentally resistant to antileishmanial drugs. *Int J Parasitol Drugs Drug Resist*, 8(2), 246-264. doi:10.1016/j.ijpddr.2018.04.002
- Silva, N., Camacho, N., Figarella, K., & Ponte-Sucre, A. (2004). Cell differentiation and infectivity of Leishmania mexicana are inhibited in a strain resistant to an ABC-transporter blocker. *Parasitology*, 128(Pt 6), 629-634. doi:10.1017/s0031182004005098
- Verma, A., Bhandari, V., Deep, D. K., Sundar, S., Dujardin, J. C., Singh, R., & Salotra, P. (2017). Transcriptome profiling identifies genes/pathways associated with experimental resistance to paromomycin in Leishmania donovani. *Int J Parasitol Drugs Drug Resist*, 7(3), 370-377. doi:10.1016/j.ijpddr.2017.10.004

Reviewer comments, further round review

Reviewer #3 (Remarks to the Author):

Guarnizo et al. have included in this revised version of their manuscript a sizable number of new experimental data. Reviewer 1 asked for the inclusion of transcriptomic data of total mRNAs and the authors performed this work. The new data supported the hypothesis of the authors that the transcriptome exhibits many differences between sensitive and resistant parasites. Reviewer 2 was laudatory with no specific comments. I was reviewer 3 with 5 initial specific comments.

1. AQP1 deletion, decreased expression, indels inducing frameshifts, are usually observed in resistant parasites and these are consistent with experimental work. I nonetheless accept the arguments of the authors that in some published work AQP1 expression was higher in resistant parasites. I had suspected that my request of studying more than one line would not be popular. Yet this is standard practice in the field and I still believe that this is important. Nonetheless I accept the argument of the authors that this would delay considerably the interesting new research direction they are proposing on translational control in drug resistance in *Leishmania*.
2. The authors have carried out extensive work on characterizing their parasite line. They showed conclusively that the resistant parasite had not a major growth defect. This is important. Note that I never said that the use of clones would be problematic. To the contrary, this is standard practice in the field, although work with populations is also reported. They have shown that their resistant parasites do not infect well macrophages. This is an issue but at least now they have documented it. I clearly do not agree with the author's definition of resistance as 10-fold the EC50. This is not what we observed with clinical isolates derived from patients failing antimony therapy.
3. I accept the arguments of the authors.
4. I accept the arguments of the authors.
5. I still do not get their concept of preemptive adaptation. They have generated a mutant resistant to a drug. So of course, these parasites are "ready" for a drug challenge. We will have to agree on disagreeing on this specific item.

REVIEWERS' COMMENTS

Reviewer #3 (Remarks to the Author):

Guarnizo et al. have included in this revised version of their manuscript a sizable number of new experimental data. Reviewer 1 asked for the inclusion of transcriptomic data of total mRNAs and the authors performed this work. The new data supported the hypothesis of the authors that the transcriptome exhibits many differences between sensitive and resistant parasites. Reviewer 2 was laudatory with no specific comments. I was reviewer 3 with 5 initial specific comments.

Response to the reviewer:

Thank you very much for the critical reading of our manuscript and providing comments and suggestions.

1. AQP1 deletion, decreased expression, indels inducing frameshifts, are usually observed in resistant parasites and these are consistent with experimental work. I nonetheless accept the arguments of the authors that in some published work AQP1 expression was higher in resistant parasites. I had suspected that my request of studying more than one line would not be popular. Yet this is standard practice in the field and I still believe that this is important. Nonetheless I accept the argument of the authors that this would delay considerably the interesting new research direction they are proposing on translational control in drug resistance in Leishmania.

Response to the reviewer:

We really appreciate your constructive criticism and productive discussion regarding multiple lines use. Thank you for accepting our argument that including more lines would substantially delay publishing of our work. We really appreciate it. You made us think deeply about our future directions. In the future studies we plan not only include more lines of *L. tropica* but also examine other species such as *L. major* and *L. donovani* to see the generality of the mechanism and reveal if other species have unique differences.

2. The authors have carried out extensive work on characterizing their parasite line. They showed conclusively that the resistant parasite had not a major growth defect. This is important. Note that I never said that the use of clones would be problematic. To the contrary, this is standard practice in the field, although work with populations is also reported. They have shown that their resistant parasites do not infect well macrophages. This is an issue but at least now they have documented it. I clearly do not agree with the author's definition of resistance as 10-fold the EC50. This is not what we observed with clinical isolates derived from patients failing antimony therapy.

Thank you for pointing this out. The degree of response to antimonial drug varies greatly between *Leishmania* species and even among strains of the same species (see Mandal et al., 2015 doi:10.1371/journal.pntd.0003500). Currently, there is no established standard what fold of change is required to recognize a strain as a drug-resistant strain. Therefore, we removed the sentence about definition of the drug resistance as 10-fold the EC50 to avoid confusion.

3. I accept the arguments of the authors.

4. I accept the arguments of the authors.

5. I still do not get their concept of preemptive adaptation. They have generated a mutant resistant to a drug. So of course, these parasites are "ready" for a drug challenge. We will have to agree on disagreeing on this specific item.

Response:

Preemptive adaptations represent parasite adjustments to the expected threat. They equip parasites with the capability to instantly provide the effective response. In this sense genotypic/phenotypic changes that occur during the development of the drug resistance to acquire the multiple adaptations that stay in place in the absence of the drug and support the parasite survival when it is exposed to the drug. This specific term is used in our work to distinguish changes of the sensitive and drug resistant strains revealed in the absence of drug, from changes the drug resistant strain exhibits in the presence of drug. We thank the Reviewer for expressing the alternative point of view. This area of research is relatively new, and the presence of different opinions are acceptable at this stage of the field development. We hope that when this area of research progresses further, the specific terminology and concepts will evolve and will be commonly used.